# Activity-induced Ca²⁺ signaling in perisynaptic Schwann cells of the early postnatal mouse is mediated by P2Y₁ receptors and regulates muscle fatigue

Dante J Heredia, Cheng-Yuan Feng, Grant W Hennig, Robert B Renden, Thomas W Gould*

Department of Physiology and Cell Biology, University of Nevada School of Medicine, Reno, United States

**Abstract** Perisynaptic glial cells respond to neural activity by increasing cytosolic calcium, but the significance of this pathway is unclear. Terminal/perisynaptic Schwann cells (TPSCs) are a perisynaptic glial cell at the neuromuscular junction that respond to nerve-derived substances such as acetylcholine and purines. Here, we provide genetic evidence that activity-induced calcium accumulation in neonatal TPSCs is mediated exclusively by one subtype of metabotropic purinergic receptor. In *P2ry1* mutant mice lacking these responses, postsynaptic, rather than presynaptic, function was altered in response to nerve stimulation. This impairment was correlated with a greater susceptibility to activity-induced muscle fatigue. Interestingly, fatigue in *P2ry1* mutants was more greatly exacerbated by exposure to high potassium than in control mice. High potassium itself increased cytosolic levels of calcium in TPSCs, a response which was also reduced *P2ry1* mutants. These results suggest that activity-induced calcium responses in TPSCs regulate postsynaptic function and muscle fatigue by regulating perisynaptic potassium.
DOI: https://doi.org/10.7554/eLife.30839.001

**\*For correspondence:**
tgould@medicine.nevada.edu

**Competing interests:** The authors declare that no competing interests exist.

## Introduction

Muscle fatigue is defined as the decline in muscle performance that occurs in response to continued muscle activation. Muscle fatigue is a clinically important feature of myopathies such as muscular dystrophy, neuromuscular disorders such as Guillain-Barré syndrome, diseases of the central nervous system (CNS) such as multiple sclerosis, or diffuse conditions such as chronic fatigue syndrome and cachexia (*Katirji, 2002*). A wide variety of mechanisms in the central and peripheral nervous systems contribute to muscle fatigue. In simplified preparations of muscle and peripheral nerve, central sources of input are eliminated, permitting the examination of peripheral sites of fatigue, such as the presynaptic release of the neurotransmitter acetylcholine (ACh) at the neuromuscular junction (NMJ; *Nanou et al., 2016*), sensitivity of postsynaptic ACh receptors, propagation of the muscle action potential along the sarcolemma and into the t-tubules, release of calcium (Ca²⁺) from sarcoplasmic reticulum, and activation of the contractile apparatus (*Boyas and Guével, 2011*). Proposed mediators of fatigue at these sites include changes in the concentration of intracellular and extracellular ions, such as calcium (Ca²⁺), sodium (Na⁺), potassium (K⁺), or protons (H⁺); metabolites, such as inorganic phosphate (P$_i$), or lactate; and reactive oxygen species (*Allen et al., 2008*). For example, high-frequency stimulation (HFS) of nerve or muscle raises the level of extracellular K⁺ or [K⁺]$_o$, which may mediate fatigue by depolarizing muscle membrane, inactivating Na$_v$1.4 voltage-gated sodium channels at the NMJ, and consequently blocking the production of muscle action potentials (APs; *Cairns et al., 2015*) This mechanism may also underlie the muscle weakness observed in patients

**eLife digest** A muscle that contracts over and over again will become tired. This can sometimes occur after vigorous exercise, but abnormal muscle fatigue is also a feature of various clinical disorders. These include conditions that affect muscles directly, such as muscular dystrophy, as well as disorders of the motor nerves that control muscles, such as Guillain-Barré syndrome.

Nerves make contact with muscles at specialized sites called neuromuscular junctions. Failing to send the correct signals to the muscles at these junctions can lead to muscle fatigue. Studies to date have focused on the role of nerve cells and muscle cells in these communication failures. But there is also a third cell type present at the neuromuscular junction, known as the terminal/perisynaptic Schwann cell (TPSC). Stimulating motor nerves in a way that produces muscle fatigue also activates TPSCs.

To investigate whether TPSCs contribute to or counteract muscle fatigue, Heredia et al. studied the responses of these cells at the neuromuscular junctions of young mice. Stimulating motor nerves caused TPSCs to release calcium ions from their internal calcium stores. However, this did not occur in mice that lacked a protein called the $P2Y_1$ receptor. In normal mice, activating the $P2Y_1$ receptor directly also made the TPSCs release calcium. This calcium release in turn prompted the TPSCs to take up potassium ions. Nerve and muscle cells release potassium during intense activity, and removal of potassium by TPSCs helped to prevent muscle fatigue.

Therapeutic strategies that make TPSCs release more of their internal calcium stores – and thus increase their potassium uptake – could help ease muscle fatigue. A valuable first step would be to use drugs and genetic techniques to show this effect in mice. The results could then guide the development of corresponding strategies in patients.

DOI: https://doi.org/10.7554/eLife.30839.002

with hyperkalemic periodic paralysis, a neuromuscular disorder caused by dominant mutations in the *Scna4* gene encoding the $Na_v1.4$ channel and characterized by episodic muscle stiffness and weakness (*Cannon, 2015*).

In addition to presynaptic nerve terminals and muscle endplates, terminal or perisynaptic Schwann cells (TPSCs) reside at the NMJ. TPSCs are a non-myelinating Schwann cell subtype that influence the regeneration of injured peripheral motor axons (*Son et al., 1996*), maintain developing synapses (*Reddy et al., 2003*), and participate in synaptic pruning (*Smith et al., 2013*). TPSCs also respond to neural activity by increasing cytosolic $Ca^{2+}$ levels (*Jahromi et al., 1992*; *Reist and Smith, 1992*) and are therefore functionally similar to other perisynaptic glial cells, such as astrocytes and enteric glia. In addition to responding to neurotransmitter released during neural activity by mobilizing $Ca^{2+}$, astrocytes regulate the concentration of extracellular metabolites produced by activity through the expression of various ion channels and transporters (*Olsen et al., 2015*; *Boscia et al., 2016*; *Weller et al., 2016*). Therefore, TPSCs, as the perisynaptic glia of the NMJ, likely act to modulate the concentrations of these ions at the NMJ and thereby regulate muscle fatigue.

In astrocytes, activity-induced $Ca^{2+}$ signaling is largely mediated by neurotransmitter-mediated stimulation of Gq G-protein coupled receptors (GPCRs), leading to the release of $Ca^{2+}$ from the endoplasmic reticulum (ER) through the second messenger inositol-1,4,5-triphosphate ($IP_3$; *Volterra et al., 2014*). Astrocytic $Ca^{2+}$ signaling in turn modulates synaptic transmission and contributes to functional hyperemia, although each of these effects remains controversial (*Agulhon et al., 2010*; *Bonder and McCarthy, 2014*). The interpretation of these effects is complicated by the observation that the mechanisms contributing to activity-induced $Ca^{2+}$ signaling in the fine processes of astrocytes are distinct from those underlying this signal in the cell body (*Bazargani and Attwell, 2016*). Additionally, the diversity of astrocyte subtypes in the brain (*Eugenín León et al., 2016*) and of neuronal subtypes associated with individual astrocytes (*Perea and Araque, 2005*) further challenge the precise identification of activity-induced $Ca^{2+}$ responses in these cells. Finally, the extent to which $IP_3R$-mediated $Ca^{2+}$ signaling reflects all of the effects of activity-induced Gq GPCR activation remain unclear (*Agulhon et al., 2013*).

TPSCs, by contrast, are only associated with the nerve terminals of cholinergic motor neurons (MNs). The NMJ is large and amenable to optical analysis owing to its discrete location at the central

endplate band region of muscle. TPSCs do not elaborate extensive processes or make specialized contacts with the microcirculation. Together, these features make the TPSC suitable for the genetic manipulation of their response to and regulation of neural activity. However, the examination of activity-induced Ca$^{2+}$ responses in TPSCs and the functional effects of these responses have been largely conducted by imaging individual TPSCs injected with fluorescent Ca$^{2+}$-binding dyes (*Darabid et al., 2014*). Thus, whether the effects of single TPSC manipulation on individual synapses lead to global effects on neuromuscular function cannot be examined.

Using mice that express the genetically-encoded calcium indicator GCaMP3 in all Schwann cells including TPSCs, we observed that high-frequency, motor nerve stimulation-induced Ca$^{2+}$ signaling within TPSCs of the neonatal diaphragm was completely abolished in the absence of the purinergic 2Y$_1$ receptor (P2Y$_1$R). We therefore utilized *P2ry1* mutant mice lacking these receptors as a model to investigate the functional effects of activity-induced, Gq GPCR-mediated Ca$^{2+}$ release in TPSCs.

## Results

### Activity-induced Ca$^{2+}$ transients in populations of TPSCs in Wnt1-GCaMP3 mice

In order to study the Ca$^{2+}$ response to neural activity in populations of TPSCs, we first evaluated transgene expression in TPSCs of the diaphragm muscle at postnatal day 7 (P7) from Wnt1-Cre, conditional GCaMP3 (Wnt1-GCaMP3) mice. Wnt1-Cre mice drive Cre-dependent transgene expression in neural crest derivatives, which include Schwann cells (*Danielian et al., 1998*). We first assessed Wnt1-Cre mice by crossing them to mice conditionally expressing the fluorescent reporter TdTomato. Robust expression of TdTomato was observed at early ages in all Schwann cells in the diaphragm, including myelinating Schwann cells of the phrenic nerve as well as non-myelinating TPSCs at the NMJ (*Figure 1A*), visualized using fluorescent α−bungarotoxin (α-BTX). We then crossed Wnt1-Cre mice to mice conditionally expressing GCaMP3, which encodes a green fluorescent protein (GFP)-conjugated calmodulin that fluoresces upon Ca$^{2+}$ binding (*Zariwala et al., 2012*). In whole-mounts of P7 diaphragm muscle incubated with GFP antibodies to label GCaMP3, S100 antibodies to detect Schwann cells, and α-BTX to visualize NMJs, we observed expression of GCaMP3 in all TPSCs (*Figure 1B*). Together, these results show that Wnt1-Cre drives robust expression of GCaMP3 in TPSCs of the early postnatal diaphragm.

We next determined if GCaMP3 expression in TPSCs exhibited activity-induced Ca$^{2+}$ responses, similar to previous studies (*Jahromi et al., 1992*; *Reist and Smith, 1992*). Imaging these responses before and after nerve stimulation at low magnification (20X), we observed large populations of TPSCs that responded to 45 s of 40 Hz tonic phrenic nerve stimulation (*Figure 2A*; *Figure 2—video 1*). Higher magnification images (60X) showed that each individual TPSC, identified by labeling with fluorescent α-BTX (data not shown), responded to HFS (*Figure 2B*). We used stat maps of the standard deviation of fluorescence intensity (SD map) to spatially represent the distribution of Ca$^{2+}$ transients within individual TPSCs from high-magnification videos and traces of intensity to examine their temporal characteristics (*Figure 2C*). The onset of these transients occurred 1.8 ± 0.74 s ($n$ = 5, $c$ = 54) after the initiation of HFS, reached peak intensity at 2.86 ± 0.49 s after the onset of the transient, and shortly thereafter declined in amplitude (time to 50% decay = 13.35 ± 4.22 s). Most transients lasted the entire duration of the nerve stimulation period, although the intensity at the end of the stimulation period was usually less than 10% of that at the initial peak. GCaMP3-expressing TPSCs responded to multiple stimulations, providing a useful tool to measure the effects of different stimuli on the same TPSCs. Similar to a previous study (*Darabid et al., 2013*), the peak intensity of Ca$^{2+}$ transients observed after a subsequent 45 s bout of 40 Hz stimulation was lower than that of the first (17.3 ± 2.5 vs. 13.3 ± 1.1 dB, first vs. second stim, $n$ = 5; p<0.05). Interestingly, analyses of transients in TPSCs across large regions of the diaphragm from low-magnification videos showed substantial variability in the onset after nerve stimulation (from 0.8 s to 4.3 s). This pattern could be achieved by differential transmitter release, differential transmitter breakdown in the perisynaptic space, or differential signal transduction in TPSCs. Because individual muscle fibers across the entire diaphragm exhibit shortening at nearly the same onset after nerve stimulation (*Figure 2—video 1*), this result does not appear to result from differential transmitter release.

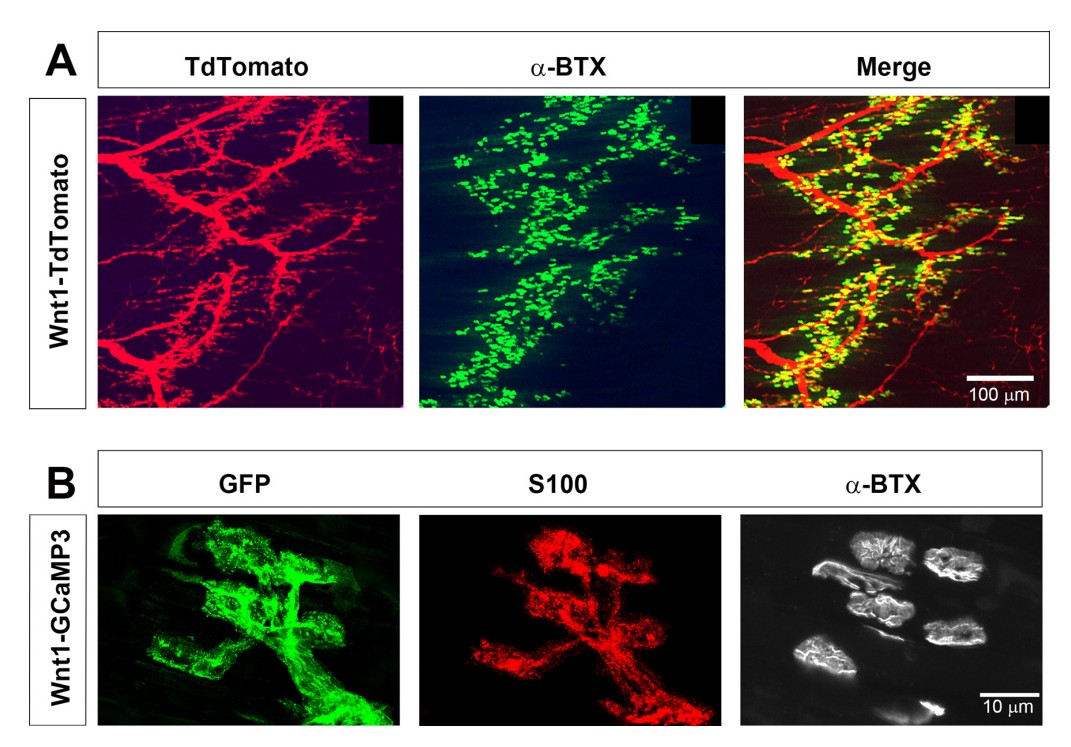

**Figure 1.** Wnt1-Cre drives expression of reporters and activity sensors to Schwann cells of the neonatal diaphragm. (A) Whole mounts of P7 Wnt1-TdTomato diaphragm were labeled with 488-conjugated α-bungarotoxin (α-BTX). Myelinating Schwann cells along phrenic nerve branches as well as terminal/perisynaptic Schwann cells at α−BTX-labeled neuromuscular junctions (NMJs; green) exhibit tdTomato epifluorescence (red). (B) Higher magnification of whole mounts of P7 Wnt1-GCaMP3 diaphragm labeled with GFP, S100, and 633-conjugated α-BTX. All NMJ-associated, S100-immunostained TPSCs express GFP and thus GCaMP3.

DOI: https://doi.org/10.7554/eLife.30839.003

Although the diaphragm is activated by short bursts of tonic stimulation of the phrenic nerve during several behaviors (e.g., expulsive maneuvers such as wretching; *Hodges and Gandevia, 2000*), the more native physiological pattern of stimulation that occurs during respiration is phasic with a duty cycle between 25% and 50%, in which each period of activation is 100 ms, at frequencies between 30–70 Hz (*Kong and Berger, 1986*; *Zhan et al., 1998*; *van Lunteren and Moyer, 2003*; *Sieck et al., 2012*). The duration of TPSC $Ca^{2+}$ transients was longer in response to 45 s of 40 Hz phasic vs. tonic stimulation (*Figure 2E,F*). The off-period of each duty cycle could clearly be discerned as a dropoff in transient intensity (*Figure 2E*), showing the dynamic nature of these responses (*Todd et al., 2010*; *Darabid et al., 2013*). At lower frequencies of stimulation, we observed similar differences of $Ca^{2+}$ signals in response to phasic vs. tonic patterns. Interestingly, in response to 10 Hz stimulation, the lowest phasic rate that produced a measurable response, $Ca^{2+}$ transients were much slower in onset after nerve stimulation than after 40 Hz stimulation (*Figure 2F*).

We next examined whether lower frequencies were capable of inducing $Ca^{2+}$ transients in TPSCs, similar to astrocytes (*Sun et al., 2014*). Whereas spontaneous ACh release, 1 Hz and 2 Hz evoked stimulation failed to produce visible $Ca^{2+}$ transients in TPSCs, 5 Hz stimulation elicited transients in a small number of TPSCs, and 10–40 Hz stimulation produced responses in all TPSCs (data not shown). In response to different durations of 40 Hz stimulation, only several TPSCs responded to 0.1 s of nerve stimulation (i.e., four pulses), whereas all TPSCs responded to 1–30 s of 40 Hz stimulation (*Figure 2—video 2*). Finally, in contrast to TPSCs, myelinating Schwann cells along phrenic nerve trunks and branches failed to exhibit $Ca^{2+}$ responses, similar to previous reports (*Jahromi et al., 1992*). However, bath application of adenosine triphosphate (ATP) or muscarine elicited a response in Schwann cells along distal phrenic nerve branches as well as TPSCs (*Figure 2—video 3*). Despite

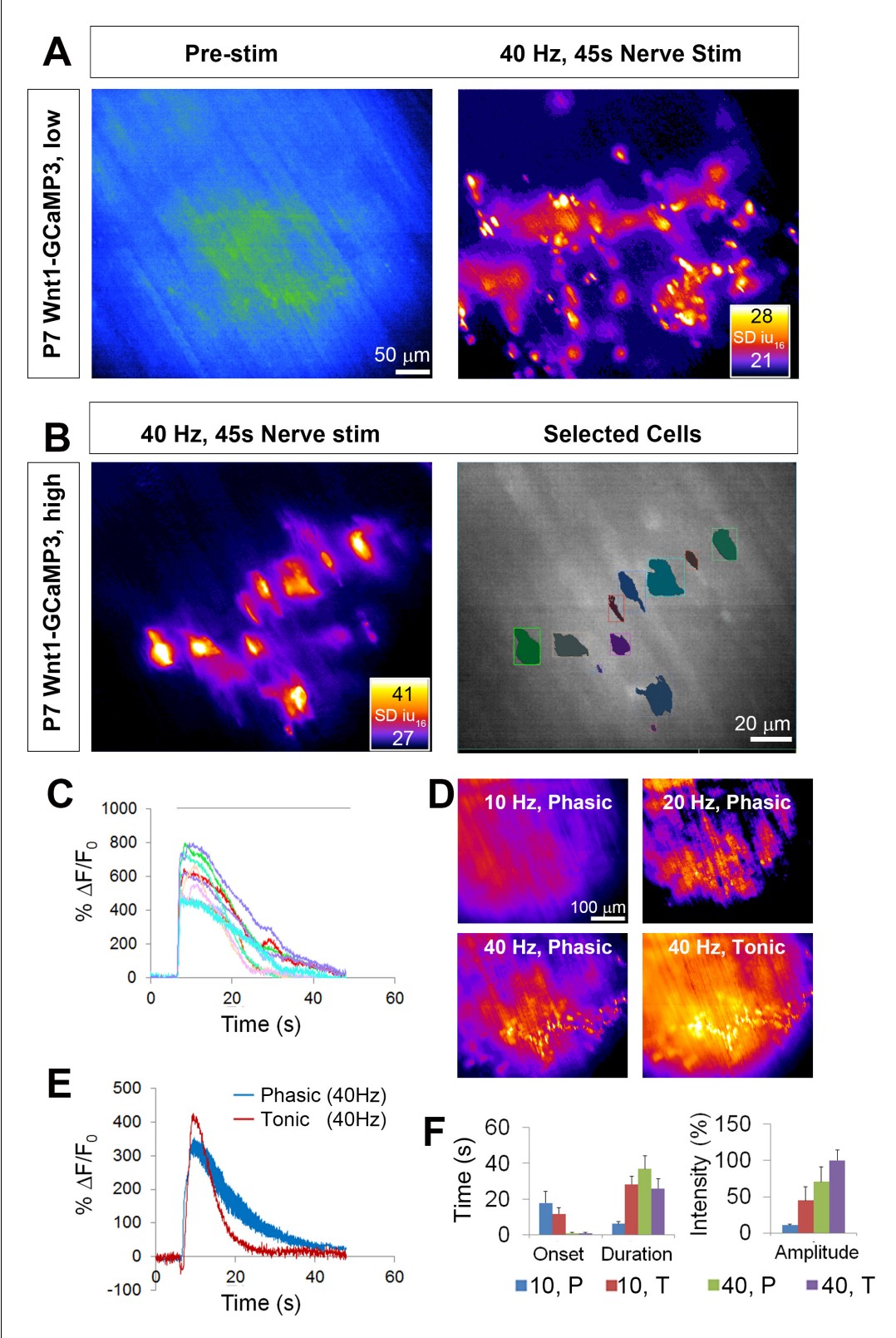

**Figure 2.** Wnt1-GCaMP3 mice exhibit activity-induced Ca$^{2+}$responses in all perisynaptic glia cells of the neonatal diaphragm. (**A**) (Left panel) An average intensity image generated before application of a stimulus (Pre-stim) shows the overall structure of GCaMP3-expressing Schwann cell elements. (Right panel) Map of standard deviation of 16-bit fluorescence intensity units (SD iu$_{16}$) of a population of TPSCs imaged in response to high-frequency nerve stimulation at low magnification; fire CLUT heatmap in SD iu$_{16}$. (**B**) Same muscle imaged at higher power showing these fluorescence responses

*Figure 2 continued on next page*

Figure 2 continued

in individual TPSCs (left panel) were color coded (right panel). (C) These cells in B were plotted as color-coded transients. Note the relatively higher signals in this graph, compared to other graphs in this study, as a result of imaging at 60X, vs. 20X in others. (D) SD maps resulting from tonic vs. phasic HFS. (E) Transients elicited by 40 Hz tonic (red) or (blue) phasic stimulation. Peak transient intensities were statistically non-significant (14.7 ± 0.4 vs. 13.5 ± 1.3 dB; p=0.21; $n = 3$; $c = 18$), and duration was longer (time to 50% decay = 15.4 ± 0.7 vs. 22.3 ± 3.7 s; p<0.05) in response to phasic vs. tonic stimulation. (F) Slower onsets and lower amplitudes of $Ca^{2+}$ transients in TPSCs stimulated with 10 vs. 40 Hz stimulation (10 Hz phasic onset = 17.92 ± 6.3; 10 Hz tonic onset = 11.78 ± 3.4 s; 40 Hz phasic onset = 0.99 ± 0.34 s; 40 Hz tonic onset = 1.0 ± 0.38 s; $p<5.3*10^{-19}$, 1-way ANOVA; every individual comparison significant except 40 Hz phasic onset vs. 40 Hz tonic onset, based on q, $q_{crit}$ comparisons). 10 = 10 Hz; 40 = 40 Hz; T = tonic; p=phasic.

DOI: https://doi.org/10.7554/eLife.30839.004

The following video and source data are available for figure 2:

**Source data 1.** These are fluorescence values of calcium transients of individual TPSCs taken at 60X (from P7 Wnt1-GCaMP3 mice), as depicted by the boxes in the right panel of *Figure 2B*, in response to 45 s of 40 Hz phrenic nerve stimulation, in the presence of the muscle-specific myosin inhibitor BHC.
DOI: https://doi.org/10.7554/eLife.30839.005

**Source data 2.** These are fluorescence values of calcium transients of individual TPSCs at P7 taken at 20X in response to 45 s of 40 Hz tonic or phasic phrenic nerve stimulation.
DOI: https://doi.org/10.7554/eLife.30839.006

**Source data 3.** Mean values of the intensity of P7 TPSC calcium transients, in decibels, in response to 45 s of 10 Hz or 40 Hz phasic or tonic phrenic nerve stimulation, were collected and represented as % TPSC calcium transient in response to 45 s of 40 Hz tonic nerve stimulation.
DOI: https://doi.org/10.7554/eLife.30839.007

**Figure 2—video 1.** Perisynaptic glia at the neuromuscular junction exhibit activity-induced $Ca^{2+}$signals in response to fatiguing, high-frequency nerve stimulation (HFS).
DOI: https://doi.org/10.7554/eLife.30839.008

**Figure 2—video 2.** Perisynaptic glia at the neuromuscular junction exhibit activity-induced $Ca^{2+}$signals in response to fatiguing levels of HFS even before the onset of fatigue.
DOI: https://doi.org/10.7554/eLife.30839.009

**Figure 2—video 3.** Schwann cells along distal axon branches, in addition to perisynaptic Schwann cells at the neuromuscular junction, exhibit $Ca^{2+}$-signals in response bath application of ATP.
DOI: https://doi.org/10.7554/eLife.30839.010

their expression of GCaMP3 (determined by immunohistochemistry; data not shown), myelinating Schwann cells along larger phrenic nerve branches failed to respond to either nerve stimulation or bath application of ATP or muscarine. Together, these results show that TPSCs dynamically respond to different patterns of neuronal activity.

## Activity-induced $Ca^{2+}$ responses in TPSCs of the neonatal diaphragm are mediated exclusively by purinergic stimulation of $P2Y_1$ receptors

Previous studies of individual or small cohorts of TPSCs show that a variety of neurotransmitter-derived substances trigger cytosolic $Ca^{2+}$ accumulation, including ACh through muscarinic AChRs, adenine nucleotides such as ATP/ADP through purinergic P2Y receptors (P2YR), and adenosine, derived from synaptic ectonucleotidase-mediated degradation of adenine nucleotides (*Cunha et al., 1996*), through P1R (*Darabid et al., 2014*). A recent study provided evidence that TPSCs also respond to nerve-derived ACh through nicotinic AChRs (*Petrov et al., 2014*). In contrast to these responses in adult TPSCs, a recent report demonstarted that TPSC $Ca^{2+}$ signals in neonatal mouse soleus are not mediated by muscarinic or nicotinic ACh receptor (mAChR) or by adenosine receptor activation, but rather by P2YRs (*Darabid et al., 2013*). In agreement with this finding, we found that whereas the pan-muscarinic antagonist atropine and the pan-nicotinic antagonist curare failed to block activity-mediated $Ca^{2+}$ transients in TPSCs of the P7 diaphragm ($n = 8$), the wide spectrum P2 antagonist suramin completely eliminated them ($n = 6$; *Figure 3C*). We tested whether this response was mediated by $P2Y_1$ receptors ($P2Y_1Rs$), as TPSCs reportedly express this protein (*Darabid et al., 2013*) and astrocytic $Ca^{2+}$ signaling is mediated in part by the Gq GPCR-coupled pathway that is activated by $P2Y_1Rs$ (*Fam et al., 2000*). Treatment with the selective $P2Y_1R$ antagonist MRS2500 (1 µM) completely blocked activity-induced $Ca^{2+}$ responses in all TPSCs of the diaphragm ($n = 5$; *Figure 3C*). In order to determine whether activity-dependent, $P2Y_1R$-mediated $Ca^{2+}$ responses were dependent on release from intracellular stores, we examined them in the presence of the sarco-/endoplasmic reticulum $Ca^{2+}$-ATPase (SERCA) inhibitor cyclopiazonic acid (CPA). 15 min after

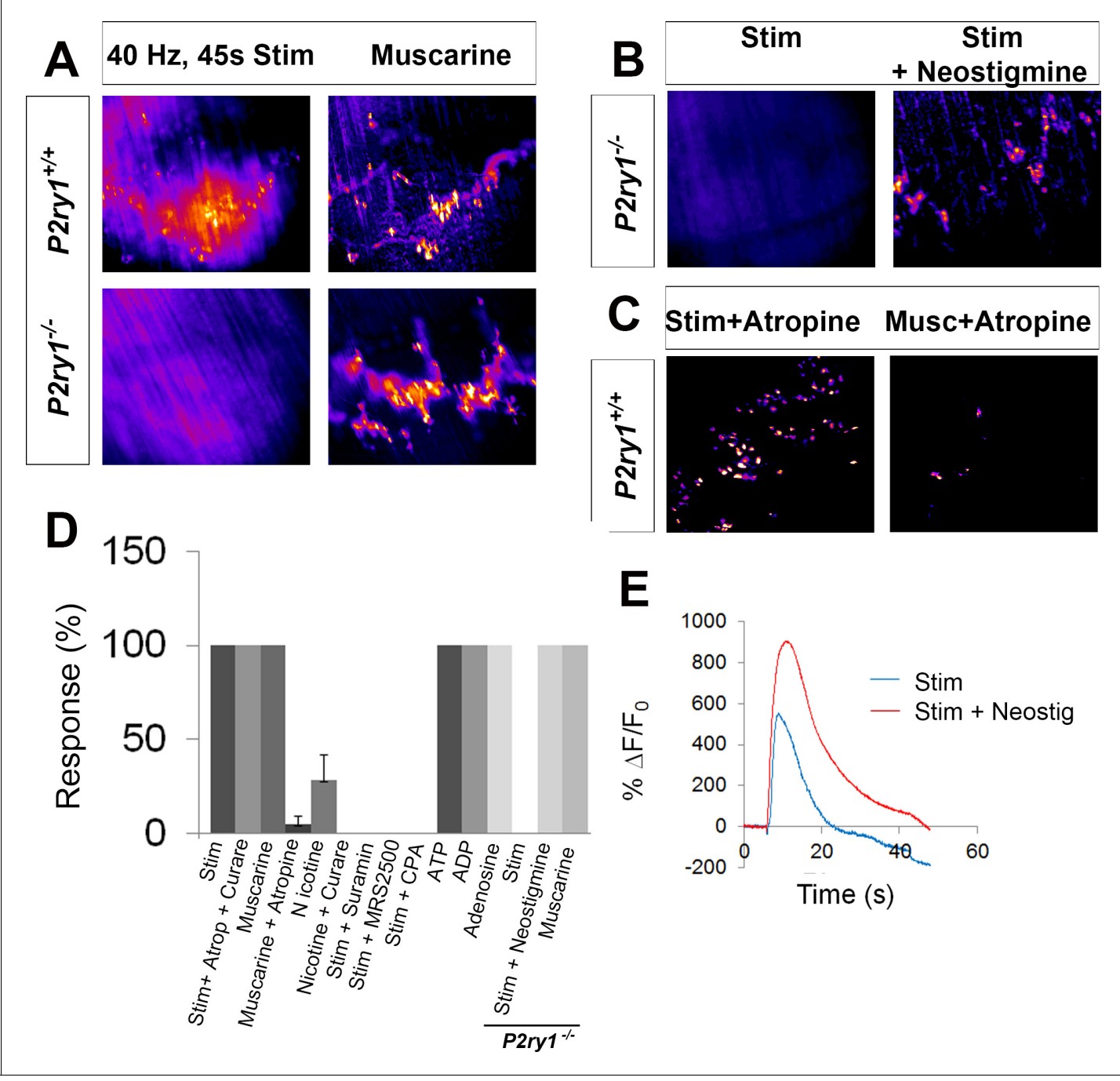

**Figure 3.** Activity-induced responses in perisynaptic glia at the NMJ are completely dependent on P2Y$_1$R signaling. (**A**) SD maps of activity-induced (left panels) or muscarine-induced (right panels) Ca$^{2+}$ responses in TPSCs of *P2ry1* wild-type (WT; top panels) or mutant (bottom panels) diaphragm. Note the complete absence of activity-induced Ca$^{2+}$ responses in TPSCs of *P2ry1* mutants; fire CLUT heatmap in SD iu$_{16}$. (**B**) Addition of neostigmine restores this response. (**C**) Atropine largely blocks muscarine-induced, but not nerve stimulation-induced, TPSC Ca$^{2+}$ responses. (**D**) Graph representing the relative number of α-BTX-associated TPSCs exhibiting Ca$^{2+}$ responses in response to activity or drug treatment. $p < 5.3 \times 10^{-19}$, 1-way ANOVA; Nicotine treatment induced a variable response (28.3 + 13.3% of stim-activated cells activated by nicotine; variance = 176.3). (**E**) WT mice exhibit elevated Ca$^{2+}$ responses to activity in the presence of the cholinesterase-blocking drug neostigmine (peak TPSC Ca$^{2+}$ intensities, 16.3 ± 1.4 vs. 20 ± 2.3 dB, stim vs. stim + neostigmine, $p < 0.05$; $n = 4$; $c = 22$).

DOI: https://doi.org/10.7554/eLife.30839.011

The following video, source data, and figure supplements are available for figure 3:

*Figure 3 continued on next page*

# eLIFE Research article

Neuroscience

*Figure 3 continued*

**Source data 1.** The number of P7 TPSCs responding (displaying a calcium transient) to each of the conditions were collected and represented as the percent of TPSCs responding to 45 s of 40 Hz phrenic nerve stimulation.

DOI: https://doi.org/10.7554/eLife.30839.014

**Source data 2.** These are fluorescence values of calcium transients of individual TPSCs from P7 WT mice, taken at 20X in response to 45 s of 40 Hz tonic phrenic nerve stimulation, in the presence of absence of the wide spectrum cholinesterase inhibitor neostigmine.

DOI: https://doi.org/10.7554/eLife.30839.015

**Source data 3.** These are the diameters in square microns of synaptophysin-immunoreactive presynaptic terminals of P7 *P2ry1* WT and mutant mice, shown in *Figure 3—figure supplement 1*.

DOI: https://doi.org/10.7554/eLife.30839.016

**Source data 4.** These are the depths in microns of the junctional folds of the postsynaptic muscle membrane of P7 *P2ry1* WT and mutant mice, shown in *Figure 3—figure supplement 2*.

DOI: https://doi.org/10.7554/eLife.30839.017

**Figure supplement 1.** Normal expression of pre-, peri- and post-synaptic elements at the NMJ of *P2ry1* mutant mice lacking activity-induced $Ca^{2+}$ responses in perisynaptic glia.

DOI: https://doi.org/10.7554/eLife.30839.012

**Figure supplement 2.** Normal ultrastructural appearance of the NMJ of *P2ry1* mutant mice lacking activity-induced $Ca^{2+}$ responses in perisynaptic glia.

DOI: https://doi.org/10.7554/eLife.30839.013

**Figure 3—video 1.** Activity-induced $Ca^{2+}$ responses in perisynaptic glia at the neuromuscular junction are mediated by $P2Y_1$Rs.

DOI: https://doi.org/10.7554/eLife.30839.018

---

treatment with CPA, these responses were completely abolished (*n* = 3; *Figure 3C*). Finally, to test whether the effect of MRS2500 was indeed mediated by $P2Y_1$Rs, we crossed mice expressing a constitutive null mutation of the *P2ry1* gene, which encodes $P2Y_1$Rs (*Fabre et al., 1999*), to Wnt1-GCaMP3 mice. Nerve stimulation of *P2ry1* mutants completely failed to elicit $Ca^{2+}$ responses in TPSCs (*n* = 7; *Figure 3A,C*; *Figure 3—video 1*). Mutant TPSCs exhibited a robust response to muscarine (*Figure 3A,C*), indicating that the failure of activity to induce these responses was not caused by non-specific effects of the *P2ry1* mutation.

We were intrigued by the inability of atropine to block activity-mediated $Ca^{2+}$ responses in TPSCs of *P2ry1* WT mice, since (a) bath application of muscarine evoked a robust $Ca^{2+}$ signal.; (b) atropine blocked this effect of bath-applied muscarine (data not shown); (c) ACh is released upon nerve stimulation. On the one hand, this may result from the absence of clustered mAChRs in P7 TPSCs (*Darabid et al., 2013*). Alternatively, the lateral diffusion of nerve-derived ACh to perisynaptic TPSC-derived mAChRs may be limited by the activity of the cholinesterases acetylcholinesterase (AChE) or butyrylcholinesterase (BChE) in the synaptic cleft. Nerve stimulation of *P2ry1* mutants in the presence of the pan-cholinesterase inhibitor neostigmine resulted in a robust $Ca^{2+}$ response in all TPSCs, suggesting that cholinesterase activity normally prevents this effect (*n* = 3; *Figure 3B,C*). Neostigmine also increased the intensity of activity-induced $Ca^{2+}$ transients in *P2ry1* WT mice, demonstrating the additive nature of the effects of purinergic and muscarinic stimulation on this response (*Figure 3D*).

We investigated which nerve-derived, $P2Y_1$R-stimulating ligands were capable of evoking TPSC $Ca^{2+}$ responses. While bath application of either ATP or ADP induced these responses, neither ADP-ribose nor β-nicotinamide adenine dinucleotide (βNAD) did (*n* = 4; data not shown; *Mutafova-Yambolieva and Durnin, 2014*). Bath application of the P1R-activating ligand adenosine also evoked robust $Ca^{2+}$ transients in TPSCs (*n* = 3; data not shown), similar to previous studies (*Robitaille, 1995*; *Castonguay and Robitaille, 2002*). However, the onset of this response was markedly delayed, compared to that triggered by purines or ACh mimetics (ATP = 3.2 ± 1.6 s, adenosine = 15.6 ± 3.4 s; p<0.001). Together, these pharmacological and genetic studies suggest that activity-induced $Ca^{2+}$ signaling in neonatal diaphragm TPSCs is mediated by adenine nucleotide-mediated stimulation of $P2Y_1$R. This result thus permits the evaluation of the functional role of this signal.

## Activity-induced, $P2Y_1$R-mediated $Ca^{2+}$ responses in TPSCs are not required for synapse formation

In order to test whether $P2Y_1$R deletion itself or activity-induced $P2Y_1$R-mediated $Ca^{2+}$ signaling exerted gross effects on synaptogenesis of the NMJ, we first assessed the structure of NMJs by

immunohistochemical and ultrastructural techniques. We examined the tripartite NMJ by staining whole-mounts of P7 diaphragm with antibodies against synaptophysin (nerve terminals), GFP (GCaMP3-expressing perisynaptic Schwann cells) and α-BTX (postsynaptic AChR clusters). We found no difference in the total number of NMJs, the size of NMJs, the percentage of innervated NMJs, or the apposition of perisynaptic Schwann cells in *P2ry1* mutant or WT mice (*Figure 3—figure supplement 1*; data not shown). Next, we examined NMJs for AChE immunoreactivity, as previous reports of adult NMJs in *P2ry1* mutants showed a reduction in the level of expression of this cholinesterase (*Xu et al., 2015*). Using a highly specific antibody that fails to detect expression of this enzyme in *AChE* mutant mice, we were unable to observe any difference in its expression or synaptic localization (*Figure 3—figure supplement 1*). Finally, we examined NMJs by electron microscopy, and found no obvious structural abnormalities (*Figure 3—figure supplement 2*). Together, these results suggest that the gross morphological development of the NMJ, at least until P7 in the diaphragm, is unaffected in mice lacking P2Y$_1$R and activity-induced Ca$^{2+}$ signaling.

## Activity-induced, P2Y$_1$R-mediated Ca$^{2+}$ responses in TPSCs are required for postsynaptic function in response to HFS

We next evaluated the effect of eliminating activity-induced Ca$^{2+}$ signaling on the presynaptic release of neurotransmitter, based on results obtained in previous studies (*Robitaille, 1998*; *Castonguay and Robitaille, 2001*). The resting membrane potential (RMP) was unchanged between genotypes ($-68.5 \pm 5.2$ vs. $-68.4 \pm 3.3$ mV; WT vs. mutant; p=0.46; WT vs. mutant; $n = 3$, $c = 13$), and was not significantly different after vs. before a nerve stimulation bout between genotypes (data not shown). The frequency, amplitude, rise to peak, and time to 50% decay of miniature EPPs (mEPPs) were also unchanged (resting frequency: $0.38 \pm 0.15$ vs. $0.46 \pm 0.14$ events/s; p=0.11; post-stimulation frequency: $1.57 \pm 0.8$ vs. $2.01 \pm 0.8$ events/s; p=0.15; amplitude: $2.52 \pm 0.5$ vs. $2.49 \pm 0.6$ mV; p=0.44; rise to peak: $3.76 \pm 1.8$ vs. $3.69 \pm 2.7$ ms; p=0.46; time to 50% decay: $5.83 \pm 2.4$ vs. $4.81 \pm 1.8$ ms; p=0.08; WT vs. mutant; $n = 3$, $c = 13$–$19$). Individual nerve-evoked EPPs, recorded in the presence of μ-conotoxin, were also similar between *P2ry1* WT and mutant mice (*Figure 4A*). In response to HFS, EPP amplitudes at the end of the period were also similar in each genotype. These results demonstrate that basal and HFS-induced ACh release are not affected in the absence of activity-induced Ca$^{2+}$ signaling in TPSCs (*Figure 4A,D*). These results also corroborate the finding that AChE expression at the NMJ was unaffected in *P2ry1* mutants, as the durations of mEPPs and EPPs were unaffected, whereas they are longer in the absence of this enzyme (*Adler et al., 2011*).

In order to assess postsynaptic function, we took advantage of BHC, a drug which blocks contraction of skeletal muscle without affecting neurotransmission and thus allows the electrophysiological and optical evaluation of muscle APs (*Heredia et al., 2016*). We first assessed individual nerve-evoked muscle APs and observed no differences between *P2ry1* WT and mutant mice (*Figure 4B*). In order to determine the effects of HFS on muscle APs, we initially examined neural transmission failure, or the failure to transmit a successful EPP into a muscle AP, by identifying the time at which less than half of the nerve stimuli were transduced into successful muscle APs. Similar to the adult diaphragm (*Heredia et al., 2016*), P7 diaphragm exhibited multiple muscle AP profiles in response to HFS, characterized by the occurrence of failed or subthreshold APs at different timepoints after stimulation, likely reflecting differential fatiguability. We were unable to detect differences in the time to neural transmission failure in any subtype (*Figure 4D*), consistent with the failure to detect differences in EPP or mEPP amplitude, which reflect the presynaptic release of and the postsynaptic response to ACh, respectively. However, when we examined the features of *successfully* transmitted muscle APs at different stages of HFS, we found that the amplitudes were smaller and durations longer in *P2ry1* mutant relative to WT mice (*Figure 4E*). These results suggest that the muscle AP itself, rather than the transmission of the nerve impulse to the muscle, is affected in the absence of activity-induced, P2Y$_1$R-mediated Ca$^{2+}$ responses in TPSCs.

## Activity-induced, P2Y$_1$R-mediated Ca$^{2+}$ responses in TPSCs are required for the maintenance of muscle force in response to HFS

Because previous studies reported that impaired muscle APs are correlated with muscle fatigue (*Juel, 1988*), we evaluated muscle force in the P7 diaphragm of *P2ry1* mutant and WT mice. We used an optical measure of fiber shortening in whole diaphragm to measure muscle peak force and

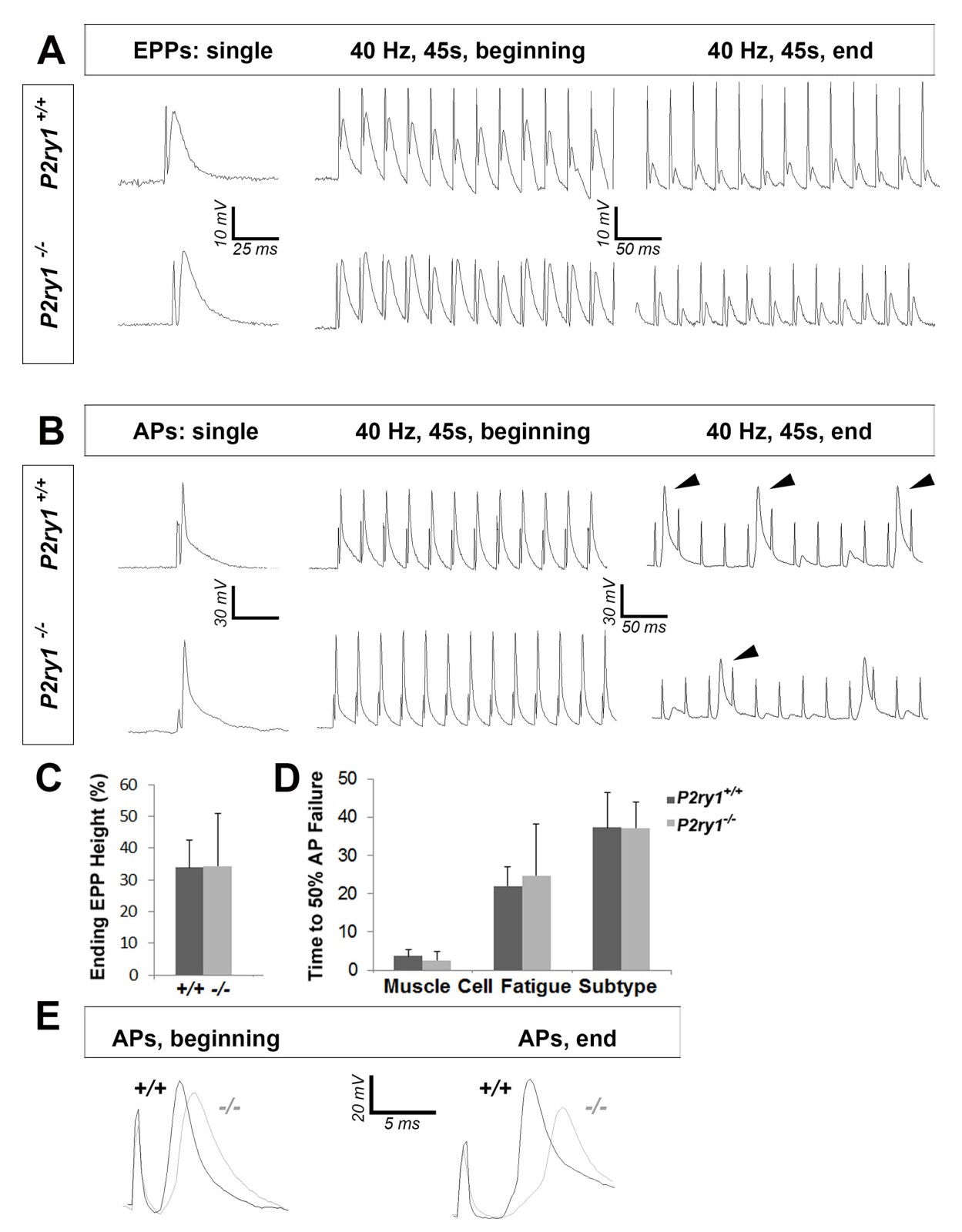

**Figure 4.** The loss of activity-induced Ca²⁺ responses in perisynaptic glia disrupts postsynaptic but not presynaptic function at the NMJ. (A) Phrenic nerve-evoked endplate potentials (EPPs) were measured in P7 diaphragm muscle from *P2ry1* WT and mutant mice in response to basal (left panel) and high-frequency stimulation (HFS; middle and right panels). There was no difference in the amplitudes (24 ± 2.6 vs. 26.7 ± 1.5 mV; p=0.13; WT vs. mutant; n = 4; c = 16) of basal EPPs or in the amplitudes of EPPs at the end of a period of HFS. Up and downward deflections preceding each EPP are

*Figure 4 continued on next page*

*Figure 4 continued*

stimulation artifacts. (B) Phrenic nerve-evoked muscle action potentials (APs) were measured in P7 diaphragm muscle from *P2ry1* WT and mutant mice in response to basal (left panel) and HFS (middle and right panels). There was no difference in the amplitudes ($67.7 \pm 7.1$ vs. $67.8 \pm 7.1$ mV; p=0.49), the rise to peak ($1.16 \pm 0.2$ vs. $1.07 \pm 0.2$ ms; p=0.33), or the time to 50% decay ($2.42 \pm 0.53$ vs. $2.16 \pm 0.82$ ms; p=0.43) of basal APs, or in the percentage of successfully transmitted muscle APs at the end of a period of HFS (note the three successful APs in the WT and 2 APs in the mutant, at the end of HFS; arrowheads). (C) As shown in right panel of A, ending EPP heights are similar between genotypes ($34 \pm 8.6$ vs. $34.3 \pm 16.6\%$ initial EPP; p=0.96; WT vs. mutant; $n = 4$; $c = 20$). (D) As shown in right panel of B, the time to 50% failure in response to HFS is similar between genotypes, in all subtypes of fatiguability ($3.7 \pm 1.6$ vs. $2.5 \pm 2.4$ s for quick fatiguability; p=0.23; $22 \pm 15.1$ vs. $24.7 \pm 13.5$ s for intermediate fatiguability; p=0.67; $37.3 \pm 9$ vs. $37.2 \pm 6.8$ s for slow fatiguability; p=0.75; WT vs. mutant; $n = 4$; $c = 27$). (E) Muscle APs from the beginning (left) or end of a train of HFS from *P2ry1* WT (black; +/+) or mutant (gray; -/-) mice. Muscle AP rise-to-peak was lengthened ($2.0 \pm 0.6$ vs. $2.7 \pm 0.7$ ms; WT vs. mutant, p<0.05) and amplitude was reduced ($58.1 \pm 3.23$ vs. $53.7 \pm 4.5$ mV; WT vs. mutant, $n = 4$, $c = 13$; p<0.05) at the end of a train of HFS.

DOI: https://doi.org/10.7554/eLife.30839.019

The following source data is available for figure 4:

**Source data 1.** These are the amplitudes of intracellularly recorded muscle endplate potentials (EPPs), relative to initial EPP amplitudes, in %, at the end of a 45 s, 40 Hz train of phrenic nerve stimulation (each value represents the average of at least 3 EPPs for that particular cell, and each animal has 4–5 cells).

DOI: https://doi.org/10.7554/eLife.30839.020

**Source data 2.** – These values represent the time at which different muscle cell types exhibit neural transmission failure, as measured by the time at which the number of successfully transmitted muscle action potentials (APs) dropped below 50% in response to 45 s of 40 Hz phrenic nerve stimulation.

DOI: https://doi.org/10.7554/eLife.30839.021

muscle fatigue (*Heredia et al., 2016*). When we examined the effect of 45 s of 40 Hz phrenic nerve stimulation, we detected no difference in the magnitude of peak contraction (*Figure 5A,B*). However, peak contraction was maintained for longer durations in *P2ry1* WT than mutant mice (*Figure 5B*). These results were also obtained in WT mice treated with the P2Y$_1$R antagonist MRS2500, demonstrating that the acute inactivation of P2Y$_1$R function is sufficient to enhance fatigue (*Figure 5C*). When we subjected the diaphragm to multiple bouts of HFS, each separated by a recovery period of 15 min, we found that the initial peak contraction, as well as ability to maintain peak contraction, were significantly reduced in *P2ry1* mutants (*Figure 5D*). Finally, in order to test whether nerve stimulation-induced muscle fatigue was enhanced as a result of impaired neuromuscular synapse transmission, we stimulated muscle rather than nerve. We failed to observe differences in peak contraction or fatigue in response to 40 s of 45 Hz electrical field between *P2ry1* WT and mutant mice (data not shown). Collectively, these data demonstrate that the perisynaptic region of muscle fibers of the P7 diaphragm is more sensitive to fatigue induced by HFS in *P2ry1* mutant than WT mice.

We next examined muscle fiber subtype in the diaphragm, since muscle AP failure profiles with different fatiguability were observed in response to HFS, and since the development of these subtypes reflects the endogenous pattern of nerve stimulation. Although earlier studies indicate that the development of these fiber subtypes first occurs at around P25 in rodent diaphragm (*Zhan et al., 1998*), we observed that both *P2ry1* WT and mutant mice contained all four basic fiber subtypes at P7, as assessed by immunostaining with myosin heavy chain (MHC) antibodies that selectively recognize each fiber subtype (*Bloemberg and Quadrilatero, 2012*). However, the relative percentage of each of these subtypes was indistinguishable between genotypes, suggesting that the enhanced fatigue in *P2ry1* mutants is not caused by a relative increase in fast-fatiguing subtypes of muscle fibers (*Figure 5E*).

A recent study demonstrated that neonatal TPSCs respond to distinct levels of nerve activity during the period of polyneuronal synapse elimination by modulating the magnitude of their Ca$^{2+}$ response (*Darabid et al., 2013*). Together with the finding that TPSC processes separate competing nerve terminals from each other and from the postsynaptic muscle fiber during this period (*Smith et al., 2013*), these data suggest that TPSC Ca$^{2+}$ responses may regulate this phenomenon. Further support for this idea comes from the finding that synapse elimination in the CNS is impaired in *P2ry1* mutant mice as well as in mice lacking activity-induced Ca$^{2+}$ responses in astrocytes (*Yang et al., 2016*). In order to evaluate this hypothesis, we confirmed that Ca$^{2+}$ responses were eliminated in the absence of P2Y$_1$R signaling at P15, the age at which synapse elimination is largely complete. Similar to those at P7, TPSC Ca$^{2+}$ responses at P15 were completely dependent on P2Y$_1$R

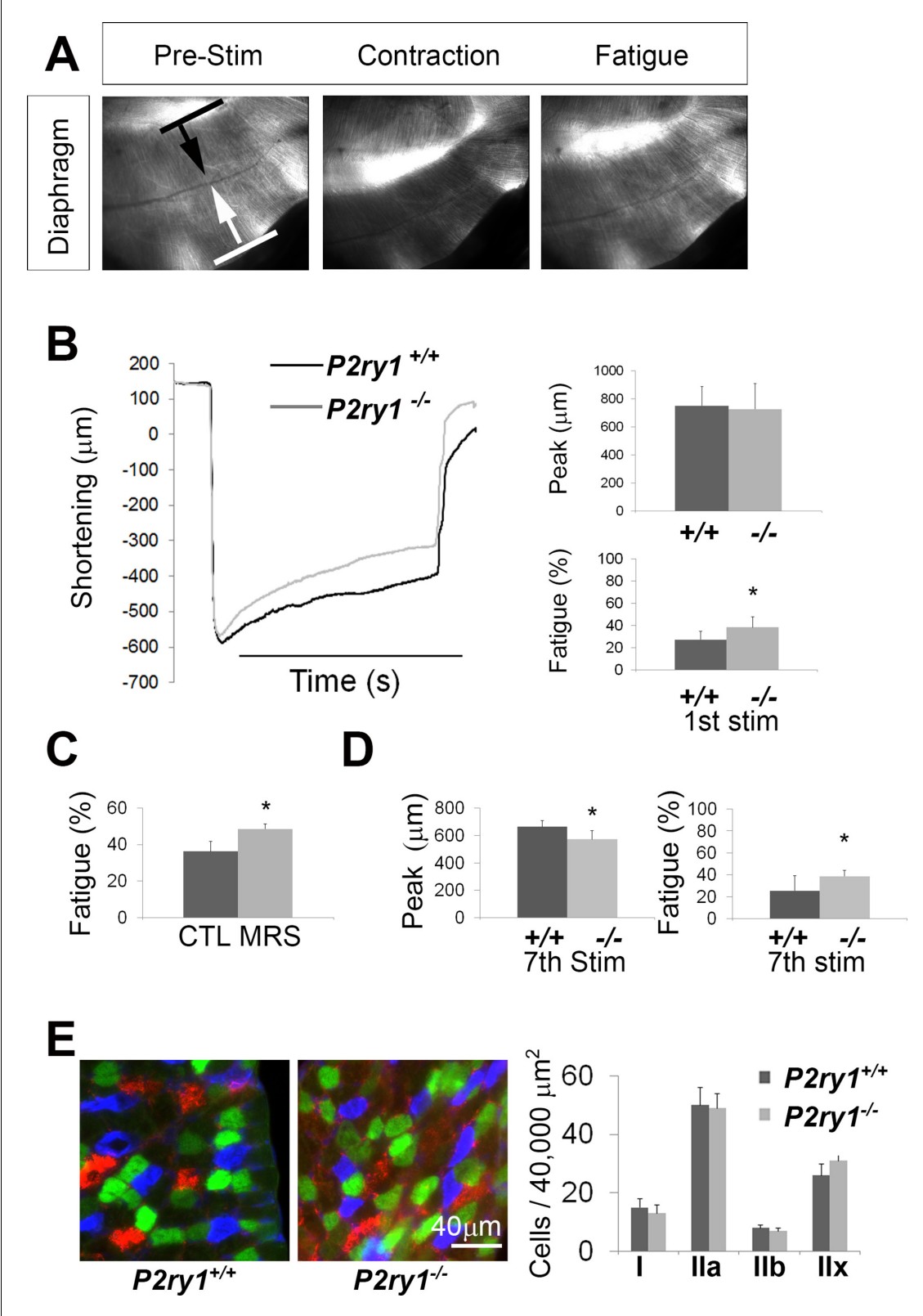

**Figure 5.** The loss of activity-induced $Ca^{2+}$ responses in perisynaptic glia leads to enhanced muscle fatigue. (**A**) Images of a P7 hemi-diaphragm before phrenic nerve stimulation, at peak contraction, and during fatigue. Black and white arrows indicate sites used to compare fiber length changes. (**B**) The shortening of muscle fibers, as measured by the change in distance between the two sites indicated by arrows in A, is represented as a negative number. Peak length changes (shortening) are similar between genotypes (751.6 ± 136 µm vs. 726 ± 182 µm; p=0.75), but are maintained significantly

*Figure 5 continued on next page*

Figure 5 continued

less over time in *P2ry1* mutants (ending length change = 72.6 ± 7.3 vs. 61.5 ± 9.5% peak length change; *n* = 9; *P2ry1* WT vs. mutant; *p<0.05). (**C**) Fatigue is also enhanced by acute blockade of TPSC $Ca^{2+}$ responses with the P2Y$_1$R antagonist MRS2500 (ending length change = 63.7 ± 10.8 vs. 51.4 ± 5.3% peak length change; *n* = 4; untreated vs. MRS2500-treated; *p<0.05; data in graph presented as the failure to maintain peak shortening or 100% minus these values). (**D**) Peak length changes in response to the seventh bout of HFS are reduced in *P2ry1* mutants (664.6 ± 43.7 μm vs. 564.4 ± 64.1 μm; *p<0.05), as is fatigue (ending length change = 74.5 ± 13.6 vs. 61.3 ± 5.8% peak length change; *n* = 4; *P2ry1* WT vs. mutant; *p<0.05). (**E**) Image of transverse section of P7 diaphragm from *P2ry1* WT and mutant mice, stained with antibodies against myosin heavy chain (MHC) Type I (blue), MHC Type IIA (green) and MHC Type IIB (red) antibodies (left panel). No difference between genotypes was observed in the number of each MHC muscle fiber subtype (15 ± 3 vs. 13 ± 3, p=0.46; Type I; 50 ± 6 vs. 49 ± 5, p=0.83; Type IIa; 8 ± 2 vs. 7 ± 2, p=0.59; Type IIb; 26 ± 4 vs. 31 ± 5, p=0.36; Type IIx; all values are *P2ry1* WT vs. mutant; *n* = 3).

DOI: https://doi.org/10.7554/eLife.30839.022

The following source data and figure supplement are available for figure 5:

**Source data 1.** These muscle shortening and fatigue curves were taken from brightfield videos of hemi-diaphragms of P7 *P2ry1* WT and mutant mice subjected to 45 s of 40 Hz phrenic nerve stimulation.

DOI: https://doi.org/10.7554/eLife.30839.024

**Source data 2.** These values for ending contraction, relative to peak contraction (fatigue) were taken from brightfield videos of hemi-diaphragms of P7 *P2ry1* WT control or MRS2500-treated mice, subjected to 45 s of 40 Hz phrenic nerve stimulation in *Figure 5C*.

DOI: https://doi.org/10.7554/eLife.30839.025

**Source data 3.** These values for peak contraction and ending contraction, relative to peak contraction (fatigue) were taken from brightfield videos of hemi-diaphragms of P7 *P2ry1* WT and mutant mice, subjected to 45 s of 40 Hz phrenic nerve stimulation for seven consecutive episodes.

DOI: https://doi.org/10.7554/eLife.30839.026

**Source data 4.** These are the numbers of different muscle fiber subtypes in images of diaphragm muscle from P7 *P2ry1* WT and mutant mice, based on myosin heavy chain immunostaining.

DOI: https://doi.org/10.7554/eLife.30839.027

**Source data 5.** These are fluorescence values of muscle calcium transients (from P7 Myf5-GCaMP3 mice) taken at 20X in response to 45 s of 40 Hz phrenic nerve stimulation for two episodes.

DOI: https://doi.org/10.7554/eLife.30839.028

**Source data 6.** These are fluorescence values of muscle calcium transients (from P7 Myf5-GCaMP3 mice) taken at 20X in response to either 45 s of 40 Hz phrenic nerve or treatment with ADP, represented as SD map images in *Figure 5—figure supplement 1C*.

DOI: https://doi.org/10.7554/eLife.30839.029

**Figure supplement 1.** The loss of activity-induced $Ca^{2+}$ responses in perisynaptic glia leads to greater and quicker loss of peak muscle $Ca^{2+}$ transient intensity.

DOI: https://doi.org/10.7554/eLife.30839.023

signaling, both in *P2ry1* WT mice treated with MRS2500 or in *P2ry1* mutant mice (*Figure 6A,B*). Fatigue was similarly enhanced, using both optical and tension measurements (*Figure 6C*). However, when we examined NMJs by neurofilament immunohistochemistry to detect the numbers of innervating axons at individual NMJs at several ages between P7 and P15, we were unable to observe any differences (*Figure 6D,E*). Together, these results suggest that activity-induced $Ca^{2+}$ responses in TPSCs are not required for polyneuronal synapse elimination in the developing diaphragm.

## Muscle fatigue is more severely enhanced by high levels of potassium in *P2ry1* mutants lacking activity-induced, P2Y$_1$R-mediated $Ca^{2+}$ responses in TPSCs

A variety of mechanisms underlie muscle fatigue. In order to assess which of these might be affected by TPSC $Ca^{2+}$ accumulation, we initially examined intracellular $Ca^{2+}$ release within muscle cells. In order to investigate activity-induced $Ca^{2+}$ signaling in whole populations of diaphragm muscle cells, we crossed Myf5-Cre to conditional GCaMP3 mice. In unparalyzed muscle under epifluorescence, we measured both fiber length changes and $Ca^{2+}$ fluorescence intensities in response to HFS. Similar to the results obtained with brightfield recordings obtained above, MRS2500 enhanced the fatigue of a second bout of HFS relative to a first, compared to no treatment (data not shown). Peak intensities and time to 50% decay of $Ca^{2+}$ transients were significantly affected in response to HFS in the presence of MRS2500 (*Figure 5—figure supplement 1*), suggesting that events upstream or concurrent with $Ca^{2+}$ release mediate muscle fatigue caused by the absence of TPSC $Ca^{2+}$ signaling. We also used Myf5-GCaMP3 mice to test whether nerve-derived purines were capable of eliciting $Ca^{2+}$ responses in muscle cells, similar to previous results (*Choi et al., 2001*). In response to 100 μM ATP,

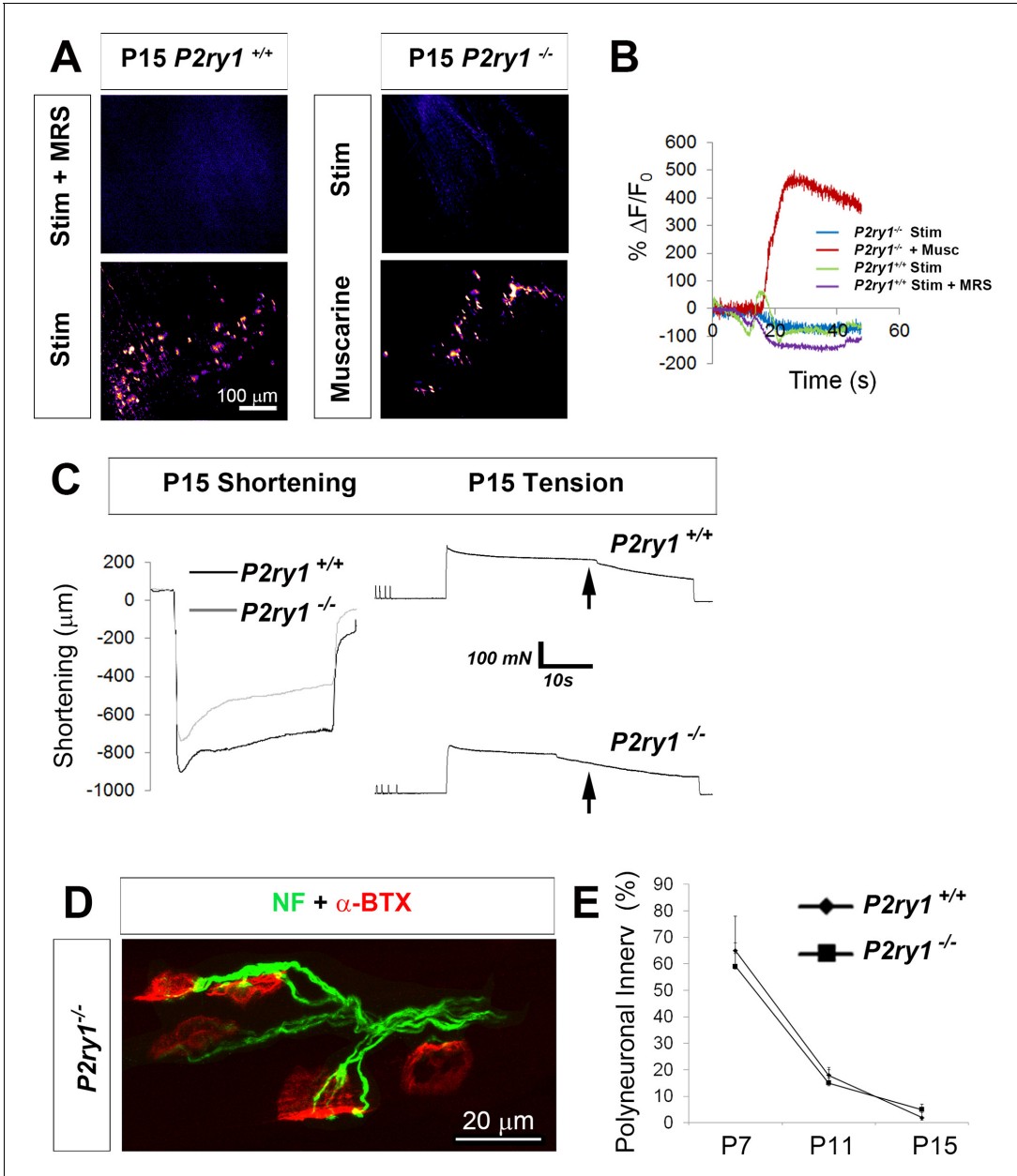

**Figure 6.** The loss of activity-induced $Ca^{2+}$ responses in perisynaptic glia does not affect the rate or magnitude of polyneuronal synapse elimination. (**A**) Pharmacological or genetic disruption of P2Y$_1$R signaling completely blocks activity-induced TPSC $Ca^{2+}$ responses at P15: *P2ry1* WT mice exhibited nerve stimulation-induced responses that were blocked after treatment with 1 μM MRS2500 (left images). *P2ry1* mutant mice failed to exhibit nerve stimulation-induced $Ca^{2+}$ responses in TPSCs but exhibited robust responses to bath-applied muscarine (right images). (**B**) Peak TPSC $Ca^{2+}$ intensities in response to these manipulations: (5.4 ± 1.2 vs. 1.5 ± 0.56 dB, WT stim vs. mutant stim, p<0.005; 5.4 ± 1.2 vs. 0.1 ± 0.04 dB, WT stim vs. WT stim +MRS2500, p<0.0001; 15.6 ± 0.7 vs. 1.5 ± 0.56 dB, mutant stim vs. mutant +muscarine, p<0.0001, Student's *t* with Bonferonni correction; c > 10 per n; n = 3–4; fluorescence units from fire CLUT heatmap in SD iu$_{16}$. (**C**) Fatigue is enhanced in P15 *P2ry1* mutants using optical measures (ending length change = 75.9 ± 8 vs. 58.6 ± 11.6% peak length change; n = 3; *P2ry1* WT vs. mutant; p<0.05) or tension recording (ending force = 39.2 ± 5.1 vs. 32.3 ± 4.3% peak force; n = 3; *P2ry1* WT vs. mutant; p<0.05). Drop in force occurs earlier in mutant vs. WT (arrows). (**D**) Polyneuronally innervated NMJs from a P7 *P2ry1* mutant diaphragm; red = AlexaFluor 594-labeled α-BTX to label AChRs; green = neurofilament immunostaining to label presynaptic nerve terminals. (**E**) The number of polyneuronally innervated NMJs is similar between *P2ry1* WT and mutant mice (P7: 65 ± 9 vs. 59 ± 13%, p=0.41; P11: 18 ± 5 vs. 15 ± 3%, p=0.34; P15: 2 ± 2 vs. 5 ± 3%, p=0.19 polyneuronally innervated NMJs; n = 5; *P2ry1* WT vs. mutant).

DOI: https://doi.org/10.7554/eLife.30839.030

The following source data is available for figure 6:

*Figure 6 continued on next page*

*Figure 6 continued*

**Source data 1.** These are fluorescence values of calcium transients of individual TPSCs from P15 *P2ry1* WT and mutant mice, or P15 *P2ry1* WT mice treated with MRS2500, taken at 20X in response to 45 s of 40 Hz phrenic nerve stimulation or in response to muscarine.
DOI: https://doi.org/10.7554/eLife.30839.031

**Source data 2.** These muscle shortening and fatigue curves were taken from brightfield videos of hemi-diaphragms of P15 *P2ry1* WT and mutant mice subjected to 45 s of 40 Hz phrenic nerve stimulation and are shown in **Figure 6C**.
DOI: https://doi.org/10.7554/eLife.30839.032

**Source data 3.** These values represent the number of multiply innervated NMJs in the diaphragm at P7, P11 and P15 in *P2ry1* WT and mutant mice and are shown in **Figure 6D**.
DOI: https://doi.org/10.7554/eLife.30839.033

muscle cells of the diaphragm of P7 Myf5-GCaMP3 mice failed to exhibit such a response (***Figure 5— figure supplement 1***).

Direct electrical stimulation of skeletal muscle produces fatigue, similar to indirect, nerve-mediated excitation. Interestingly, in response to high levels of extracellular potassium or high $[K^+]_o$, this activity-induced fatigue is enhanced, and muscle APs exhibit lower amplitudes and longer durations (***Cairns et al., 2015***), similar to the response of *P2ry1* mutants to nerve activity shown above. These results raise the possibility that activity-induced $Ca^{2+}$ signaling in TPSCs protects against muscle fatigue by regulating $K^+$ uptake by these cells and therefore persiynaptic $[K^+_o]$. similar to reports of other perisynaptic glia (***Wang et al., 2012a***, ***Wang et al., 2012b***). In order to test this idea, we first challenged diaphragms to $[K^+]_o$ greater or less than normal levels (5 mM). These challenge experiments were modeled on those used to characterize the effects of hypo- and hyperkalemia on skeletal muscle function (***Wu et al., 2011***). We found that HFS-induced fatigue was disproportionately enhanced in *P2ry1* mutant, relative to WT, mice in response to 10 mM $[K^+]_o$ (***Figure 7A***). The enhanced fatigue in these mutants was further revealed by multiple bouts of HFS; *P2ry1* mutant diaphragm was almost completely unable to contract after the second period of HFS in high $[K^+]_o$, in marked contrast to *P2ry1* WT diaphragm (***Figure 7A***; ***Figure 7—video 1***). We next examined whether the effect of high $[K^+]_o$ was caused by depolarization of the postsynaptic muscle membrane. After stimulation with several bouts of HFS in 5 mM $[K^+]_o$, muscle cells were impaled and recorded before and after changing the $[K^+]_o$ to 10 mM. While this caused a mild depolarization of the RMP in *P2ry1* WT mice, this effect was enhanced in *P2ry1* mutants (***Figure 7B***). Conversely, HFS-induced fatigue was modestly but not significantly ameliorated in response to low $[K^+]_o$ in *P2ry1* mutants (***Figure 7C***). Collectively, these results demonstrate that P7 diaphragm muscle cells lacking activity-induced, $P2Y_1R$-mediated $Ca^{2+}$ signaling in TPSCs are more sensitive to high $[K^+]_o$, suggesting that $Ca^{2+}$ signaling modulates the response to $[K^+]_o$ in these perisynaptic glia.

## *P2ry1* mutants lacking activity-induced, $P2Y_1R$-mediated $Ca^{2+}$ responses in TPSCs exhibit a reduced potassium-induced $Ca^{2+}$ response

To test if TPSCs respond to and/or regulate $K^+$ at the NMJ, we examined the response of these cells to changes of $[K^+]_o$. Interestingly, we found that raising $[K^+]_o$ to 10 mM resulted in a robust TPSC $Ca^{2+}$ response (***Figure 7D***). This was not mediated by indirect depolarizing effects of $[K^+]_o$ on the phrenic nerve, leading to an activity-induced, $P2Y_1R$-mediated $Ca^{2+}$ response, because it was still observed in the presence of doses of tetrodotoxin that blocked neurotransmission, and because it was also observed in *P2ry1* mutants completely lacking activity-induced $Ca^{2+}$ responses. In other words, these responses were observed in the absence of nerve stimulation. Interestingly, $Ca^{2+}$ responses induced by high $[K^+]_o$ were not caused by influx of extracellular $Ca^{2+}$ into TPSCs, because they were still observed when external $Ca^{2+}$ was removed, but rather by release from intracellular stores, because they were abrogated after treatment with CPA ($n$ = 3; data not shown). In order to determine if TPSCs directly respond to manipulations of $[K^+]_o$, we performed whole cell voltage recordings of these cells, identified by GCaMP3 expression in P7 Wnt1-GCaMP3 mice and co-localization with α-BTX-labeled NMJs. We observed that the RMP of TPSCs depolarized in response to treatment with 10 mM KCl by an amount that was close to that predicted by the Hodgkin-Goldman-Katz equation (data not shown). Therefore, these data demonstrate that TPSCs are capable of taking up $K^+$ and that elevations of $[K^+]_o$ depolarize TPSCs, leading to a release of $Ca^{2+}$ from intracellular

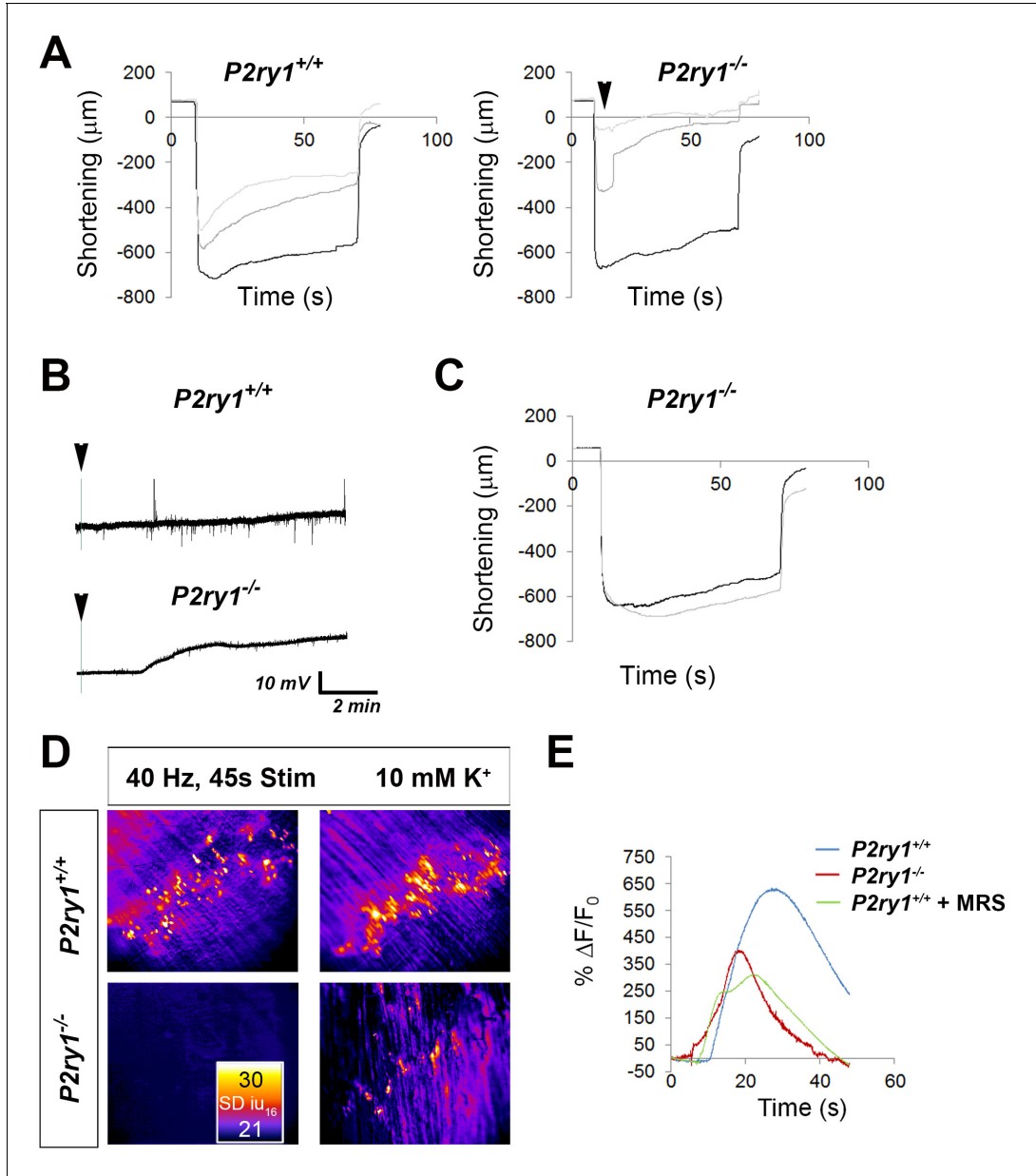

**Figure 7.** High $[K^+_o]$ triggers greater muscle fatigue, greater muscle membrane depolarization, and reduced $Ca^{2+}$ responses in the absence of activity-induced $Ca^{2+}$ responses in perisynaptic glia. (**A**) Muscle length changes from P7 diaphragm of *P2ry1* WT (left) and mutant (right) mice were first recorded in response to HFS in normal or 5 mM $[K^+_o]$ (black traces), and then recorded in response to 2 bouts of HFS (60 Hz, 45 s) in 10 mM $[K^+_o]$ (dark and light grey traces = 1st and 2nd HFS, respectively). Peak length changes and fatigue are dramatically affected in *P2ry1* mutants by high $[K^+_o]$: peak length in response to first HFS in 10 mM $[K^+_o]$ as a percentage of peak length in response to control HFS in 5 mM $[K^+_o]$ = 93 ± 20.7 vs. 53.8 ± 24%; $n = 3$; *P2ry1* WT vs. mutant mice, $p<0.05$; peak length change in response to second HFS in 10 mM $[K^+_o]$ = 74.2 ± 15.5 vs. 20.9 ± 11% control HFS; $n = 3$; *P2ry1* WT vs. mutant mice; $p<0.005$; asterisk indicates this almost complete failure to contract; fatigue, or ending length change of second HFS in high 10 mM $[K^+_o]$ = 41.3 ± 17.8 vs. 7.4 ± 6.8 peak length change of first HFS in 5 mM $[K^+_o]$; $n = 3$; *P2ry1* WT vs. mutant mice, $p<0.05$). (**B**) Muscle length changes in *P2ry1* mutants in response to HFS in lowered $[K^+_o]$ show a statistically non-significant trend toward less fatigue (ending length change, 2.5 mM $[K^+_o]$ (grey trace) vs. 5 mM $[K^+_o]$ (black trace)=75.1 ± 6.2 vs. 83.9 ± 12.9% peak length change; $n = 3$, $p=0.17$). (**C**) Effect of high $[K^+_o]$ on resting membrane potential (RMP). Representative muscle cell recording before and after (arrowhead) $[K^+_o]$ was changed from 5 to 10 mM in P7 diaphragm from *P2ry1* WT (left) and mutant (right). (**D**) SD intensity maps of TPSCs in response to HFS (left panels) in *P2ry1* WT (upper) and mutant (lower) mice and in response to subsequent treatment with 10 mM $[K^+_o]$ (right panels). Markedly fewer TPSCs responded to elevated $[K^+_o]$ in *P2ry1* mutants. (**E**) Peak $Ca^{2+}$ transient intensities of TPSCs responding to 10 mM $[K^+_o]$ after 3 bouts of HFS were significantly reduced in *P2ry1* mutants or in WT mice treated with the P2Y1R antagonist MRS2500 (18.6 ± 2.5 vs. 14.5 ± 1.5 dB, WT vs. mutant, $p<0.05$, $c > 10$ per $n$; $n = 4$, Student's $t$ with Bonferonni correction; 18.6 ± 2.5 vs. 13.8 ± 0.8 dB, WT vs. WT + MRS2500, $p<0.01$, $c > 10$ per $n$; $n = 4$, Student's $t$ with Bonferonni correction; fluorescence units from fire CLUT heatmap in SD $iu_{16}$).

*Figure 7 continued on next page*

*Figure 7 continued*

DOI: https://doi.org/10.7554/eLife.30839.034

The following video and source data are available for figure 7:

**Source data 1.** These muscle shortening and fatigue curves were taken from brightfield videos of hemi-diaphragms subjected to 45 s of 60 Hz phrenic nerve stimulation in the presence of normal or high levels of extracellular potassium in P7 *P2ry1* WT mice.

DOI: https://doi.org/10.7554/eLife.30839.035

**Source data 2.** These muscle shortening and fatigue curves were taken from brightfield videos of hemi-diaphragms subjected to 45 s of 60 Hz phrenic nerve stimulation in the presence of normal or high levels of extracellular potassium in P7 *P2ry1* mutant mice.

DOI: https://doi.org/10.7554/eLife.30839.036

**Source data 3.** These muscle shortening and fatigue curves were taken from brightfield videos of hemi-diaphragms subjected to 45 s of 60 Hz phrenic nerve stimulation in the presence of normal or low levels of extracellular potassium in P7 *P2ry1* mutant mice and shown in *Figure 7C*.

DOI: https://doi.org/10.7554/eLife.30839.037

**Source data 4.** These are fluorescence values of calcium transients of individual TPSCs from P7 *P2ry1* WT and mutant mice, or P7 *P2ry1* WT mice treated with MRS2500, taken at 20X in response to treatment with 10 mM potassium chloride.

DOI: https://doi.org/10.7554/eLife.30839.038

**Figure 7—video 1.** Treatment with high levels of potassium markedly enhances fatigue in *P2ry1* mutant mice lacking activity-induced Ca$^{2+}$ responses in perisynaptic glia at the neuromuscular junction.

DOI: https://doi.org/10.7554/eLife.30839.039

stores, similar to depolarization-induced intracellular Ca$^{2+}$ release reported in neurons (*Ryglewski et al., 2007*).

If K$^+$ uptake is affected in TPSCs of *P2ry1* mutants lacking activity-induced Ca$^{2+}$ signaling, then this [K$^+$]$_o$-induced Ca$^{2+}$ response in these cells might in turn be affected. In order to test this idea, we subjected diaphragms to several bouts of HFS, separated by 15 min, to mimic the effects on muscle fatigue described above, then assessed the effects of high [K$^+$]$_o$ on Ca$^{2+}$ signaling. In contrast to *P2ry1* WT mice, mutants or WT mice treated with MRS2500 showed a markedly reduced Ca$^{2+}$ response to 10 mM [K$^+$]$_o$ (*Figure 7D,E*), suggesting that K$^+$ uptake is impaired in *P2ry1* mutants. This failure of TPSCs to regulate perisynaptic [K$^+$]$_o$ may contribute to the enhanced muscle fatigue that occurs in *P2ry1* mutants lacking activity-induced TPSC Ca$^{2+}$ responses. Additionally, these results suggest that TPSC Ca$^{2+}$ responses, and consequently K$^+$ uptake, are positively regulated by nerve activity through feedforward (i.e., neurotransmitter-mediated stimulation) and feedback mechanisms (i.e., by suprathreshold [K$^+$]$_o$ itself.

## Discussion

Our results using Wnt1-GCaMP3 mice demonstrate that activity-induced Ca$^{2+}$ signaling in neonatal TPSCs of the diaphragm is mediated by P2Y$_1$R activation by nerve-derived adenine nucleotides. The absence of Ca$^{2+}$ signaling within TPSCs does not appear to affect the structural and molecular development of the NMJ, nor does it alter the presynaptic release of neurotransmitter, but rather affects the postsynaptic AP during sustained HFS, where longer, smaller APs were correlated with a failure to maintain peak muscle force. Because previous studies observed that administration of high K$^+$ induced similar effects on muscle APs subjected to fatiguing muscle stimulation (*Cairns et al., 2015*), we examined muscle fatigue in response to this treatment and found that it was enhanced to a greater degree in *P2ry1* mutants lacking activity-induced Ca$^{2+}$ signaling. This heightened susceptibility to high [K$^+$]$_o$ may be caused by impaired K$^+$ uptake by *P2ry1* mutant TPSCs, as these cells exhibited a markedly reduced release of Ca$^{2+}$ from intracellular stores in response to high [K$^+$]$_o$ treatment. Collectively, these results suggest that activity-induced, P2Y$_1$R-mediated Ca$^{2+}$ signaling in TPSCs influences muscle fatigue by regulating perisynaptic [K$^+$]$_o$ (*Figure 8*).

The current study represents the first evaluation of Ca$^{2+}$ responses using genetically encoded calcium indicators in perisynaptic glia at the NMJ. The onset after nerve stimulation and time to peak intensity of Ca$^{2+}$ transients using this method are similar to those of published studies using Ca$^{2+}$-sensitive dyes in TPSCs loaded with Fluo-3 as well as astrocytes expressing GCaMP3 (*Jahromi et al., 1992*; *Reist and Smith, 1992*; *Darabid et al., 2013*; *Akerboom et al., 2013*), demonstrating that this genetic technique is a valid tool to measure these responses in large populations of TPSCs. The most striking finding from this study is the complete dependence of TPSC Ca$^{2+}$ responses on a

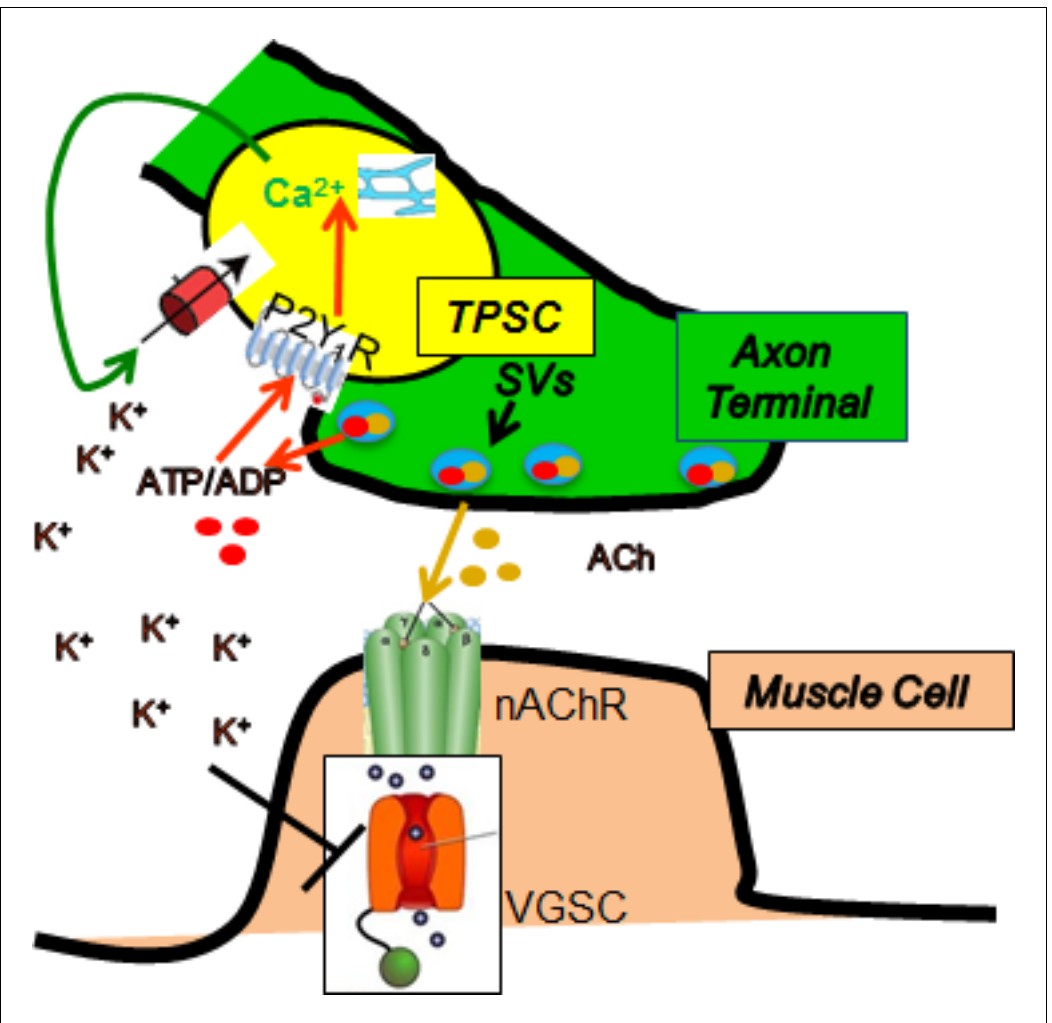

**Figure 8.** Proposed model by which nerve stimulation-induced $Ca^{2+}$ responses in TPSCs regulate muscle fatigue. Upon stimulation, presynaptic motor axon terminals (green) release acetylcholine (ACh) from synaptic vesicles (SVs), which elicits endplate potentials via postsynaptic nicotinic ACh receptors (nAChR), followed by activation of voltage-gated sodium channels (VGSC), leading to action potentials and contraction of muscle (brown). Nerve terminals also release ATP, which as itself or ADP stimulates in TPSCs (yellow) the release of calcium ($Ca^{2+}$) from intracellular stores via P2Y$_1$ receptors (P2Y$_1$R). This signal leads to the movement into TPSCs of perisynaptic potassium ($K^+$), produced by both neurons and muscle cells in response to stimulation. This regulation of perisynaptic $K^+$ levels by TPSCs is proposed to reduce the inactivation of VGSCs by $K^+$ in the neuromuscular synapse during repetitive stimulation, thus reducing muscle fatigue.
DOI: https://doi.org/10.7554/eLife.30839.040

single GPCR, P2Y$_1$R. Therefore, despite the fact that bath administration of a multiple substances induces widespread $Ca^{2+}$ signaling in neonatal TPSCs, activity-induced responses are only mediated by adenine nucleotides. In contrast to these results, studies of adult TPSCs support a role for ACh and other factors in mediating these responses (*Darabid et al., 2014*). Therefore, the early exclusive dependence on P2Y$_1$R-activating adenine nucleotides may broaden over time. Alternatively, TPSCs at the NMJs of the diaphragm may continue to depend exclusively on P2Y$_1$R signaling. At the oldest ages at which we were able to examine population responses (P15-P20), activity-induced $Ca^{2+}$ signals were completely dependent on the P2Y$_1$R pathway, supporting this latter idea. Moreover, the prevention of ACh diffusion to perisynaptic mAChRs by cholinesterase is unlikely to represent a developmentally transient response. Indeed, a recent report described a functional role for TPSC-derived BChE at the adult NMJ (*Petrov et al., 2014*).

Our results fail to support the idea that TPSC $Ca^{2+}$ responses affect the presynaptic release of ACh, in contrast to previous studies (*Robitaille, 1998*; *Castonguay and Robitaille, 2001*). NMJs at the diaphragm do exhibit dynamic changes of ACh release such as facilitation and depression (*Vautrin et al., 1993*), arguing against the absence of plasticity at diaphragm NMJs as an explanation of these differences. Rather, they may be attributable to different species, muscle, age or technique. Interestingly, using a similar genetic approach to study the effect of eliminating activity-induced $Ca^{2+}$ responses in astrocytes neurotransmitter release was unaffected (*Agulhon et al., 2010*). On the other hand, postsynaptic function was affected at the NMJ in response to HFS in *P2ry1* mutants. Whereas the number of successful muscle APs, reflecting the presynaptic release of and response to ACh, was not different in response to HFS, the characteristics of these APs changed significantly in response to prolonged nerve stimulation. These results suggest that muscle APs may not be transduced as efficiently in *P2ry1* mutant mice. The reduced intensity of muscle $Ca^{2+}$ transients in response to pharmacological blockade of $P2Y_1Rs$ supports this contention, as does the enhancement of muscle fatigue in response to pharmacological or genetic inhibition of this pathway. These effects may result from the absence of $P2Y_1R$ function in muscle, as the mice used in this study were constitutive mutants. For example, *Choi et al. (2001)* found that stimulation of chick muscle fibers with 100 µM ATP triggered $Ca^{2+}$ release from intracellular stores, a response blocked by the pan-P2 blocker suramin. Together with the expression of $P2Y_1R$ by muscle, these findings suggest that nerve-derived ATP may modulate intracellular muscle $Ca^{2+}$ levels and thus muscle fatigue. However, we failed to observe $Ca^{2+}$ release in response to ATP in the muscle cells of Myf5-GCaMP3 mice. Moreover, $Ca^{2+}$ signals in TPSCs were observed in response to treatment with lower doses of ADP/ATP (10–20 µM) than those used to evoke these signals in chick muscle cells. Finally, $Ca^{2+}$ signals in TPSCs were blocked after $P2Y_1R$ was blocked by specific pharmacological and genetic tools, rather than the pan-P2R blocker suramin (*Choi et al., 2001*). Thus, we favor the idea that the reduction of muscle cell $Ca^{2+}$ mobilization in *P2ry1* mutants is caused by the absence of this protein in TPSCs rather than in muscle cells.

The effects of HFS on muscle APs suggested the possibility that perisynaptic $[K^+]_o$ was dysregulated in *P2ry1* mutants. Supporting this idea, treatment with high $[K^+]_o$ enhanced muscle fatigue to a greater extent in *P2ry1* mutant than in WT mice. Together with a report that intense exercise increases $[K^+]_o$ in muscle to 10–14 mM (*Mohr et al., 2004*), a level sufficient to cause muscle fatigue (*Sjøgaard, 1990*); but see *Shushakov et al., 2007*), these data indicate that muscle fatigue is enhanced in *P2ry1* mutants as a result of elevated perisynaptic $[K^+]_o$, caused by a failure of TPSCs to spatially buffer or take up $K^+$ (*Kofuji and Newman, 2004*). In order to test this idea, we examined the effects of treatment with high $[K^+]_o$, reasoning that if $K^+$ uptake mechanisms in TPSCs were impaired in *P2ry1* mutants lacking activity-induced $Ca^{2+}$ signaling, these effects would be diminished. Indeed, we found that $Ca^{2+}$ responses to 10 mM $[K^+]_o$ were markedly reduced in these mutants, providing indirect evidence that $K^+$ uptake was impaired. Interestingly, elevation of $[K^+]_o$ to 20 mM also induced a robust increase of intracellular $Ca^{2+}$ in cultured astrocytes (*Duffy and MacVicar, 1994*). However, in contrast to the current study, this response was mediated entirely by external influx through voltage-gated $Ca^{2+}$ channels (VGCC). Therefore, $K^+$-induced $Ca^{2+}$ responses in TPSCs do not appear to result from depolarization-mediated ingress through VGCC, but rather by the release from intracellular stores. Together, these data suggest that activity stimulates perisynaptic $K^+$ uptake in TPSCs by both feedforward $Ca^{2+}$ responses initiated by neurotransmitter and feedback signals initiated by high $[K^+]_o$ itself.

The intracellular uptake of $K^+$ has been demonstrated in Müller glia in the retina (*Newman et al., 1984*) as well as in other glial subtypes and may be mediated by several mechanisms, including inwardly-rectifying potassium channels (Kir), $Na^+$, $K^+$ ATPases, and $Na^+$, $K^+$ $Cl^-$ cotransporters. The importance of perisynaptic $K^+$ regulation by glial cells has been demonstrated by several genetic studies. For example, mice lacking Kir4.1 in astrocytes exhibit impaired $K^+$ and neurotransmitter uptake, leading to seizures, ataxia and early lethality (*Djukic et al., 2007*). In contrast, *Wang et al. (2012b)* observed that activity-induced $Ca^{2+}$ responses in astrocytes are required for $K^+$ uptake through an ouabain-sensitive $Na^+$, $K^+$ ATPase activity. Future studies will determine which if any inward $K^+$ conductance is expressed in and stimulated by activity within TPSCs, as well as the mechanisms by which increases of intracellular $Ca^{2+}$ lead to enhanced $K^+$ uptake. Of note, it has been established that nonmyelinating Schwann cells in sympathetic nerves possess Kir currents that are sensitive to neural activity (*Konishi, 1994*).

In addition to $[K^+]_o$, the extracellular concentrations of other ions are dysregulated during muscle fatigue, including $[Na^+]_o$ and $[H^+]_o$ (*Allen et al., 2008*). Because astrocytes express an abundance of transporters and ion channels that modulate the levels of these ions in response to neuronal activity, TPSCs may similarly regulate these ions in response to stimulation of muscle, in addition to $[K^+]_o$. On the other hand, muscle itself is equipped for this role, expressing an abundance of ion channels and transporters along the extensive t-tubule system that regulate muscle membrane potential and excitability during activity (*Fraser et al., 2011*). However, the importance of ionic homeostasis at the NMJ may depend on additional mechanisms, such as those proposed here. For example, the sensitivity of $Na_v1.4$, which is expressed at high levels in a restricted region in the depths of the postsynaptic junctional folds (*Stocksley et al., 2005*), to the inactivating effects of depolarization (*Cannon, 2015*), may require enriched expression of $[K^+]_o$ buffering proteins by TPSCs (*Figure 8*). Consistent with this idea, *P2ry1* mutants did not exhibit enhanced fatigue in response to direct muscle stimulation.

In summary, we have utilized the diaphragm of neonatal *P2ry1* mutant mice as a model to explore the functional significance of activity-induced $Ca^{2+}$ signals in perisynaptic glia. We found that in the absence of purinergic signaling, postsynaptic rather than presynaptic function was altered, leading to enhanced muscle fatigue in response to HFS. These effects were correlated with elevated $[K^+]_o$ and reduced responsivity to $[K^+]_o$, suggesting that activity-induced $Ca^{2+}$ responses in TPSCs regulate perisynaptic $[K^+]_o$. Future studies will determine the mechanisms underlying $K^+$ uptake and $[K^+]_o$-mediated $Ca^{2+}$ accumulation in TPSCs. Such mechanisms may represent important translational targets in diseases with altered $[K^+]_o$. For example, in patients with hyperkalemic periodic paralysis, a genetic disorder caused by *Scna4* mutations and characterized by elevated $[K^+]_o$, stimulation of $Ca^{2+}$ signaling and subsequently $[K^+]_o$ buffering within TPSCs may enhance neuromuscular function.

## Materials and methods

**Key resources table**

| Reagent type (species) or resource | Designation | Source or reference | Identifiers |
| --- | --- | --- | --- |
| strain, strain background (mus musculus) | Wnt1-Cre | Jax | 9107 |
| strain, strain background (mus musculus) | Myf5-Cre | Jax | 7893 |
| strain, strain background (mus musculus) | conditional GCaMP3 | Jax | 14538 |
| strain, strain background (mus musculus) | conditional GCaMP6f | Jax | 24105 |
| strain, strain background (mus musculus) | conditional tdTomato | Jax | 7905 |
| strain, strain background (mus musculus) | P2ry1 mutant | Jax | 9131 |
| chemical compound, drug | BHC | Hit2lead | 5102862 |

### Ethical approval and use of mice

*P2ry1* mutant, GCaMP3 or GCaMP6f conditional knockin, and *Wnt1-Cre* and *Myf5-Cre* transgenic mice were all purchased from Jax. *P2ry1* null mutant mice were backcrossed into the C57/Bl6 strain several times before crossing to other strains, each of which is maintained in the C57/Bl6 strain. We could find no difference in any experiment between *P2ry1*$^{+/+}$ and *P2ry1*$^{+/-}$ mice, so we pooled these samples and denoted them all in the text as 'WT.' Similarly, we found no difference between male and female *P2ry1*$^{+/-}$ mice, so we pooled these samples. In order to generate *P2ry1* mutants expressing GCaMP3 in Schwann cells, we generated *Wnt1-Cre; P2ry1*$^{+/-}$ and *Rosa26-GCaMP3*$^{flox/flox}$; *P2ry1*$^{+/-}$ mice and crossed them, such that all *P2ry1* mutant and heterozygote mice expressed only one copy each of Cre and GCaMP3. We used a slightly modified common 3' WT primer to genotype *P2ry1* mutant mice (ATT TTT AGA CTC ACG ACT TTC) and the recommended primers by Jax for all other alleles. All studies were performed with animals aged postnatal day 7 and 15 (P7, P15). To verify knockouts, we performed RT-PCR on muscle-derived RNA with primers against a 300 bp fragment of *P2ry1* (5': CTG TGT CTT ATA TCC CTT TCC, 3': CTC CAT TCT GCT TGA ACT C). Animal husbandry and experiments were performed in accordance with the National Institutes of Health *Guide for the Care and Use of Laboratory Animals* and the IACUC at the University of Nevada.

## Drugs

The following reagents were used at the following concentrations: $P2Y_1R$ agonists ATP and ADP (Sigma; 10 or 20 μM); $P2Y_1R$ antagonist MRS2500 (Tocris; 1 μm); P1R agonist adenosine (Sigma; 100 μM); pan-P2 antagonist suramin (Sigma; 100 μM); pan-muscarinic agonist muscarine (Sigma; 10 μM); pan-muscarinic blocker atropine (Sigma; 10 μM); pan-nicotinic agonist nicotine (Sigma; 50 μM); pan-nicotinic antagonist curare (Sigma; 200 μM); pan-cholinesterase inhibitor neostigmine (Sigma; 1 μM); sarco-/endoplasmic reticulum $Ca^{2+}$-ATPase (SERCA) inhibitor, cyclopiazonic acid (Sigma; CPA; 10 μM); potassium chloride (Sigma; 2–10 mM); GIIIb μ-conotoxin (Peptides International; 2.3 μM); skeletal muscle myosin-blocker 3-(N-butylethanimidoyl)−4-hydroxy-2H-chromen-2-one (BHC; Hit2lead; 100 μM); 488-, 594-, 633-conjugated-α-bungarotoxin (α-BTX; Biotium; 1 μg/mL).

## Calcium imaging

The diaphragm of Wnt1-GCaMP3 mice was illuminated with a Spectra X light engine (Lumencor). In order to quantify maximal fluorescence ($F_{max}$) exhibited by GCaMP3 in Schwann cells, 30 μM CPA was added to deplete sarcoplasmic reticular $Ca^{2+}$ stores (*Heredia et al., 2016*).

Image sequences were captured using an Andor Neo sCMOS camera and a Windows-based PC using Nikon NIS Elements 4.1. Image sequences were recorded at 25 frames per second, and were exported as 8-bit TIFF files into custom-written software (Volumetry G8d; logic underlying methods in Source code file 1). A Gauss filter (3 × 3 pixel, sd = 1.0) was applied to reduce camera noise, and motion-correction routines were used to stabilize neural and Schwann cell elements in the movie (see *Hennig et al., 2015*). Changes in background fluorescence were stabilized by subtracting the average intensity near the main phrenic nerve branch. An average intensity image was generated before stimulus application ('Pre-stim' ~1 s) to quantify basal $Ca^{2+}$ levels in Schwann cells. These images are presented using a blue->green color lookup table (CLUT). This image was subtracted from the entire movie, thereby filtering out static fluorescent structures and displaying only objects that changed their intensity, i.e., $Ca^{2+}$ transients. A number of statistical maps (stat maps) were used to portray and analyze the pattern of activity-induced $Ca^{2+}$ transients in TPSCs. The main stat map type used to portray the amplitude of $Ca^{2+}$ transients in TPSCs was the standard deviation (SD) map. This map was calculated in similar fashion to the average intensity image, except the standard deviation of 16-bit fluorescence intensity units (SD $iu_{16}$) at every pixel prior to the application of the stimulus (0.5–1.0 s) extending to 60 s was calculated. Intensity SD projections (SD $iu_{16}$) were used to portray changes in $Ca^{2+}$-induced fluorescence as they are more descriptive of the overall changes in intensity during the recording; average intensity projections ($iu_{16}$) can be somewhat misleading depending on how much of the 'non-active' time before and after an event is included in the average. Intensity averages also don't describe how much the signal fluctuates. Per pixel maximum or maximum-minimum calculations can also be misleading as they include the maximum noise amplitude and/or artifacts (i.e., shot noise). SD remedies these issues, as it describes intensity fluctuations (noise or event) in standardized units and it is less sensitive to the time periods chosen to perform the projection/s. Overall, this approach isolates actively fluorescing structures more clearly than other types of projections. SD maps were color coded using a 'Fire' CLUT.

Traces of fluorescence intensity were generated from movies and presented as changes in fluorescence with respect to initial fluorescence ($\Delta F/F_{avg(prestim)}$ or $\Delta F/F_o$). Peak intensities of $Ca^{2+}$ transients were calculated as a signal-to-noise ratio (SNR) in which peak standard deviation values were divided by the prestim standard deviation value. The $\log_{10}$ of this ratio was generated and multiplied by 20 to standardize the SNR as decibels (dB). Decibels (signal-to-noise) are commonly used in signal transduction to describe the strength of a signal in terms of its 'resolvability' above background. This approach allows one to definitively characterize signals in terms of the 'noise' in the surrounding environment. We have begun using imaging decibels to more accurately portray $Ca^{2+}$ signals in relation to the surrounding environment - which essentially relates the peak amplitude of the $Ca^{2+}$ event to the standard deviation of the signal before the event (on a log scale). For more information on the logic flow underlying these routines, see source code file entitled, 'Image Analysis.'

Drugs were either bath applied in proximity to the motor endplate or pressure injected (PDES-O2DX; NPI Electronic). Drugs dissolved in DMSO were either perfused in or diluted in 1 mL of Kreb's-Ringer's before bath application, as bath application of small volumes of DMSO (~8 μl into 8 ml chamber) caused fluorescence within Wnt1-GCaMP3-expressing TPSCs. For experiments with

altered potassium, a stock solution of 3M [KCl] was added to the bath to change the concentration from 5 mM to 10 mM immediately prior to imaging.

## Electrophysiology

Diaphragms were dissected and pinned on a 6 cm Sylgard-coated dish containing oxygenated Krebs-Ringer's solution at RT according to standard procedure. Stimulation and recording of intracellular potentials were performed as described (*Heredia et al., 2016*). Briefly muscle APs were recorded after treatment with 100 μM BHC for 30 min, followed by 30 min of washing. Endplate potentials (EPPs) were recorded after treatment with the $Na_v1.4$ antagonist μ-conotoxin GIIIb (μ-CTX; 2.3 μM). Signals were amplified, digitized, recorded and analyzed as described (*Heredia et al., 2016*). The temporal dynamics of EPPs (rise to peak; time to 50% decay) were recorded as gross approximations of endplate current (EPC) kinetics changes, as has been done previously in the presence or absence of cholinesterase inhibitors (*Fatt and Katz, 1951*; *Beránek and Vyskocil, 1968*; *Kuba and Tomita, 1971*). Only muscle fibers with resting membrane potentials between −60 and −75 mV were included for analysis. Stimulation episodes of the phrenic nerve over 10 Hz were separated by 30 min rest periods to allow recovery. In order to calculate percent failure (APs) or percent transmitter release rundown (EPPs), the average of three potentials at a particular timepoint (e.g., the time at which fewer than half of nerve stimuli produced a successful muscle AP) was taken and expressed as a percent of the average of the first three potentials. For experiments with altered KCl, NaCl was adjusted accordingly to maintain the same $Cl^-$ concentration before being perfused into the dish. For recording of whole cell membrane potential, diaphragms from Wnt1-GCaMP3 mice were treated with 10 mg/ml collagenase for 30 min at 37°C. TPSCs near the surface were identified by green fluorescent GCaMP3 signal in somatic cytosol, located adjacent to red fluorescent α-BTX-stained endplates. Whole cell recording configuration was achieved using 2 μm borosilicate pipettes pulled to 5–7 mΩ tip resistance. Whole cell access was 8–20 mΩ, and leak current was <-50 pA when voltage-clamped at −60 mV command potential. Pipette internal solution contained (in mM): 97.5 K-gluconate, 32.5 KCl, 40 HEPES, 12 Na-phosphocreatine, 2 MgATP, 0.5 GTP, and 0.5 EGTA. Voltage recordings were made using current-clamp mode (I = 0) on a HEKA EPC 10 amplifier controlled by Patchmaster software. Current injection steps of 100 ms duration were given in increments between −200 to +700 pA. $[K^+]_o$ was then altered by superfusion of bath solution from 5 to 10 mM KCl to determine if a potassium conductance existed in these cells at resting membrane potential.

## Fatigue

Tension recording of muscle force in diaphragm strips (P15) or video recording of muscle shortening in hemidiaphragms (P7, P15) in response to nerve or muscle stimulation was performed as described (*Heredia et al., 2016*). For experiments with altered KCl, NaCl was adjusted accordingly to maintain the same $Cl^-$ concentration before being perfused in.

## Immunohistochemistry

Antibodies against GFP (Rockland), S100 (Dako), synaptophysin (Santa Cruz), neurofilament (Millipore) and acetylcholinesterase (kindly provided by P. Taylor, UCSD) were used at 1/1000 in PBS containing 1% triton-X and 10% fetal bovine serum to detect proteins in fixed, whole-mount diaphragms. Fluorescently-conjugated α-BTX and fasciculin-2 were added with secondary antibodies. Tissues were confocally imaged with an Olympus Fluoview 1000. For myosin heavy chain staining, muscles were fresh-frozen, cut at 16 μm, and immediately incubated without fixation in PBS with primary antibodies as described (*Heredia et al., 2016*).

## Electron microscopy

P7 mice were transcardially perfused in 1.5% glutaraldehyde, 2% paraformaldehyde in 0.1M sodium cacodylate. The costal diaphragm was dissected and incubated in fixative at 4°C overnight and then in rinse for several hours at 4°C. Samples were post-fixed in 2% osmium tetroxide, dehydrated, incubated in propylene oxide, embedded in Spurr's resin and polymerized at 60°C overnight. Ultrathin sections were cut at 90 μm and stained with uranyl acetate followed by lead citrate. Sections were photographed or digitized using a Phillips CM10 transmission electron microscope equipped with a Gatan BioScan digital imaging system.

## Statistics

Power analyses were performed using G*power 3.010 to determine the numbers of *P2ry1* wild-type (WT) and mutant mice required. For example, to determine the number of mice (*n*) and cells (*c*) to analyze for electrophysiological recordings, a power of 0.8, significance or alpha of 0.05 and effect size or Pearson's r of 3.6 was used. Thus, for these experiments, data was generated from *c* = 3 per animal or more, and from *n* = 3 or more. In this case, each *c* and each *n* are biological replicates. Differences between means were assessed by unpaired Student *t*-tests, in some cases with the Bonferonni correction for multiple comparisons, assuming equal variance, or evaluated using analysis of variance (ANOVA) with Tukey post-hoc tests. As mentioned above, a *p* value < 0.05 was considered significant. Student *t*-tests were tested for significance with two tails if the direction of the outcome was not predicted (e.g., initial comparisons of morphology, MHC isoform, synapse elimination, electrophysiology, calcium signals and shortening/fatigue between *P2ry1* WT and mutants) and with one tail if an outcome was predicted (e.g., subsequent calcium and shortening/fatigue experiments between *P2ry1* WT and mutant after treatment with MRS2500 or high/low potassium). In source data files, all reported statistical tests from text or figure legends are italicized and in red font.

## Acknowledgements

This work was supported with funds from the National Institutes of Health (NIH): GM103554 and GM110767 (to TWG); National Center for Research Resources (5P20RR018751-09) and the National Institute of General Medical Sciences (8P20 GM103513-09) to GWH.

## Additional information

### Funding

| Funder | Grant reference number | Author |
|---|---|---|
| National Center for Research Resources | P20 RR018751 | Grant W Hennig |
| National Institute of General Medical Sciences | P20 GM103513 | Grant W Hennig |
| National Institute of General Medical Sciences | P20 GM103554 | Thomas W Gould |
| National Institute of General Medical Sciences | P30 GM110767 | Thomas W Gould |

The funders had no role in study design, data collection and interpretation, or the decision to submit the work for publication.

### Author contributions

Dante J Heredia, Thomas W Gould, Data curation, Software, Validation, Investigation, Visualization, Methodology, Writing—original draft; Cheng-Yuan Feng, Data curation, Validation, Investigation, Visualization, Methodology; Grant W Hennig, Conceptualization, Software, Supervision, Funding acquisition, Investigation, Visualization, Methodology; Robert B Renden, Resources, Data curation, Software, Formal analysis, Methodology

### Author ORCIDs

Cheng-Yuan Feng http://orcid.org/0000-0002-3278-9451
Thomas W Gould http://orcid.org/0000-0003-2154-9388

### Ethics

Animal experimentation: This study was performed in strict accordance with the recommendations in the Guide for the Care and Use of Laboratory Animals of the National Institutes of Health. All of the animals were handled according to approved institutional animal care and use committee (IACUC) protocol (#00-666) of the University of Nevada School of Medicine. The University is fully accredited by AAALAC International.

**Decision letter and Author response**
Decision letter https://doi.org/10.7554/eLife.30839.047
Author response https://doi.org/10.7554/eLife.30839.048

## Additional files

### Supplementary files

• Source code 1. The logic underlying several methods used in Volumetry to generate standard deviation of fluorescence intensity map and spatio-temporal maps for TPSC Ca$^{2+}$ imaging, as well as tissue distortion using edges for muscle shortening and fatigue. Each method is broken down conceptually into several categories, including variables needed, storage buffers, and approach taken. This logic can be utilized across a wide variety of software programs.
DOI: https://doi.org/10.7554/eLife.30839.041

• Transparent reporting form
DOI: https://doi.org/10.7554/eLife.30839.042

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
