## [Decision Letter]

Thank you for submitting your article "Activity-induced Ca^2+^ signaling in perisynaptic Schwann cells is mediated by P2Y_1_ receptors and regulates muscle fatigue" for consideration by *eLife*. Your article has been reviewed by three peer reviewers, and the evaluation has been overseen by a Reviewing Editor and Richard Aldrich as the Senior Editor. The following individuals involved in review of your submission have agreed to reveal their identity: Wesley Thompson (Reviewer #1); Markus Schwab (Reviewer #2).

The reviewers have discussed the reviews with one another and the Reviewing Editor has drafted this decision to help you prepare a revised submission.

Heredia et al. have studied an important topic in glial cell biology, the role of terminal/perisynaptic Schwann cells (TPSCs) at the neuromuscular junction (NMJ) in the control of motor functions. Specifically, the authors investigated a preparation of the diaphragm muscle (which includes the phrenic nerve) from neonatal transgenic mice that express a genetically encoded Ca-sensor TPSCs, and in TPSCs lacking the purinergic P2Y_1_ receptor. The authors carefully quantify the Ca responses and show the stimulus frequencies that elicit them and the temporal characteristics of the signals. Major conclusion of the paper are (1) nerve activity-induced Ca^2+^ signaling in neonatal TPSCs is mediated by neuronal adenine nucleotide-mediated stimulation of P2Y_1_R, which is nevertheless dispensable for normal development and for synapse elimination; and (2) Ca^2+^ signaling in TPSCs modulates the transduction of muscle action potentials and thereby adds to muscle fatigue. The authors provide evidence that the latter is not synaptic but rather relates to the ability of muscle cells to generate action potentials, possibly reflecting the perisynaptic potassium concentration. This study significantly adds to our knowledge about TPSC functions in motor control and provides a conceptual framework for subsequent studies on the mechanisms of glial Ca-signaling at the NMJ.

Towards the end of the Results section, the paper becomes somewhat hard to follow and should be better explained, e.g. the potassium buffering idea. The authors should also deal with the following:

1) There is no direct measurement of potassium uptake by TPSCs shown (or the measurement of perisynaptic K^+^ by K^+^-sensitive electrodes). The most direct measurement could be the difference between the anticipated membrane potential of the glia – given a change in potassium concentration – and the actual measured membrane potential. There is also no obvious connection between external potassium manipulations and the calcium release inside the glia. Do potassium manipulations produce the glial potassium responses in the absence of nerve stimulation?

2) The calculation of the "rise to peak" or "time to 50% decay" is not appropriate for EPPs or mEPPs because of the cell's capacitance. Can the respective current responses be measured in a voltage-clamp procedure?

3) It is unclear how the TPSCs located at the synapse can affect the fatigue of the entire muscle fiber. If the presumed effect is on the portion of the muscle fiber under the synapse (and perhaps the transmission of action potentials from this region of the fiber to the rest of the fiber) this should be discussed. The authors should also address the question over what length of fiber the altered perisynaptic potassium can be effective in changing action potentials of the entire fiber.

4) Revision of Figures:

- Figure 1: are the sensory axons of the phrenic nerve recombined in Wnt-Cre mice (i.e. tdtomato+)? What exactly is depicted here (add to legend)?

- Figure 2: add significance.

- Figure 3: add variance and p values.

- Figure 4: consider changing the order (why is Figure 4 before 4C, D?);

- Figure 4: mention that the early up and downward deflections are stimulation artifacts;

- Figure 4: lacks quantification.

- Figure 6: adding 10 mM [K^+^]o had markedly different effects in muscle cells from wildtype and P2Y_1_-deficient mice. Can this be reproduced in wildtype muscle preincubated with MRS 21500?

- Figure 3—figure supplement 1 and Figure 3—figure supplement 2: the quantification of data is missing.

- Figure 6: did the authors also use P2Y_1_R mutant mice for calcium imaging at P15? It would be more consistent to include P2Y_1_R KO data in A/B. Preserved calcium signaling at P15 should be shown.

---

## [Author Response]

Towards the end of the Results section, the paper becomes somewhat hard to follow and should be better explained, e.g. the potassium buffering idea. The authors should also deal with the following:1) There is no direct measurement of potassium uptake by TPSCs shown (or the measurement of perisynaptic K^+^ by K^+^-sensitive electrodes). The most direct measurement could be the difference between the anticipated membrane potential of the glia – given a change in potassium concentration – and the actual measured membrane potential. There is also no obvious connection between external potassium manipulations and the calcium release inside the glia. Do potassium manipulations produce the glial potassium responses in the absence of nerve stimulation?

Last question first: After treating the preparation with tetrodotoxin, electrical nerve stimulation no longer produced muscle contraction. In these TTX-treated preps, treatment with potassium still produced calcium responses in TPSCs. Therefore, the calcium responses were not likely to be caused indirectly by potassium-mediated nerve activation, followed by nerve-derived, ATP-mediated P2Y_1_R activation on TPSCs. Additionally, potassium triggered TPSC calcium responses in P2Y_1_R mutants, in which nerve stimulation itself failed to elicit calcium responses. This result also suggests that the TPSC calcium response to potassium is not indirectly caused by an effect of potassium on the nerve, followed by nerve-derived ATP-mediated P2Y_1_R activation on TPSCs.

Second to last question: We do not know the mechanism by which potassium triggers calcium responses in TPSCs, other than that it is not caused by external calcium, but rather by release from internal stores. We do have new information that potassium depolarizes TPSCs (see directly below). Therefore, TPCSs, similar to neurons (Ryglewski et al., PLoS Biol, 2007 5(4):e66), may possess a membrane voltage sensor that activates metabotropic signaling pathways leading to intracellular calcium release.

First question: With the help of Robert Renden, a colleague in the department who specializes in the recording of whole cell potentials and currents in tissue slices, we tested whether TPSCs exhibited a response to external potassium ([K^+^]_o_.) by obtaining whole cell voltage recordings of perisynaptic/terminal Schwann cells (TPSCs) from P6-P7 diaphragm muscle derived from *Wnt1-GCaMP3* mice. After treatment of the muscle with 10mg/ml collagenase for 30 minutes at 37 ^0^C, TPSCs near the surface were identified by green fluorescent GCaMP3 signal in somatic cytosol, located adjacent to red fluorescent a-BTX-stained endplates (see Author response image 1). Whole cell recording configuration was achieved using 2-mm borosilicate pipettes pulled to 5-7 mΩ tip resistance. Whole cell access was 8-20 mΩ, and leak current was <-50 pA when voltage-clamped at -60 mV command potential. Pipette internal solution contained (in mM): 97.5 K-gluconate, 32.5 KCl, 40 HEPES, 12 Na-phosphocreatine, 2 MgATP, 0.5 GTP, and 0.5 EGTA. Voltage recordings were made using current-clamp mode (I=0) on a HEKA EPC 10 amplifier controlled by Patchmaster software.

Current injection steps of 100 ms duration were given in increments between -200 to +700 pA (see Author response image 2). No spiking or voltage-dependent conductance was observed in two cells, while a single small spike (<-20 mV peak) was observed in one cell. Depolarization due to +100 pA current injections was used to calculate input resistance (R=dV/dI) at 0.171 ± 0.025 mΩ. Resting membrane potential was between -42 and -30 mV (-36.6 ± 5.7mV), after correcting for 9mV liquid junction potential. [K^+^]_o_. was then altered by superfusion of bath solution from 5 to 10 mM KCl to determine if a potassium conductance existed in these cells at resting membrane potential. Hodgkin-Goldman-Katz (HGK) equation predicted changes of +4.59 mV between 5 and 10 mM KCl, in these recording conditions. Our observed changes were +4.53+ ± 0.37 mV shift in three cells. In summary:

- No spiking

- Low input resistance

- Closely predicted change, according to HGK equation, of resting membrane potential in response to changes of [K^+^]_o_.

**Author response image 2. respfig2:** 

These results suggest that two of the three cells recorded from shared characteristics with central perisynaptic glial cells, and contained a potassium-permeable channel, at rest. The identity of the third cell is unknown, but showed nonlinearity in input resistance, a small voltage-dependent spike, and less negative resting membrane potential.

We think based on their preliminary nature, together with the fact that this represents the first analysis of whole cell currents of TPSCs that we are aware of, that these data are best included as data not shown. We have included the data as such, but have included a description of the methods in the appropriate sub-section.

2) The calculation of the "rise to peak" or "time to 50% decay" is not appropriate for EPPs or mEPPs because of the cell's capacitance. Can the respective current responses be measured in a voltage-clamp procedure?

While we were aware that the EPC is faster than the EPP based on seminal contributions of Takeuchi and Takeuchi (1959), and that EPC kinetics change based on the changing potential during the EPP (Magleby and Stevens, 1972) we were still under the impression that a comparison of EPP/mEPP kinetics between two genotypes not shown/presumed to exhibit different muscle membrane capacitance was valid. The literature appears full of examples of such comparisons (Cantuti-Castelvetri et al; J Neurosci, 2015 35(4):1606-16; Yang et al., Neuroscience, 2007 149(4): 768–778, etc.) including those studying the function of TPSC Ca^2+^ signaling (Castonguay and Robitaille J Neurosci., 2001 21(6):1911-22). Of particular relevance to our manuscript, Fatt and Katz (J Physiol, 1951 115(3):320-70.); Beránek and Vyskocil (J Physiol 1968 195(2):493-503) and Kuba and Tomita (J Physiol 1971 213(3): 533–544) as well as others showed a lengthening of the decay of the EPP in response to AChE inhibitors, while many others have demonstrated similar effects examining EPCs. However, we concede that we have not empirically demonstrated that *P2ry1* wild-type and mutant muscle membrane capacitance is not different.

Our major goal was to determine the amplitude of the EPP in *P2ry1* wild-type and mutants at the beginning and end of high-frequency trains. Thus, while an analysis of the kinetics of membrane currents from muscle endplates is possible and would better support the lack of functional AChE involvement at the *P2ry1* mutant NMJ than the temporal kinetics of EPPs/mEPPs, we do not feel that such data would sufficiently enhance the main findings of our manuscript.Accordingly, if the reviewers wish, we will remove the rise-to-peak and time-to-decay data of the mEPPs.

3) It is unclear how the TPSCs located at the synapse can affect the fatigue of the entire muscle fiber. If the presumed effect is on the portion of the muscle fiber under the synapse (and perhaps the transmission of action potentials from this region of the fiber to the rest of the fiber) this should be discussed. The authors should also address the question over what length of fiber the altered perisynaptic potassium can be effective in changing action potentials of the entire fiber.

First item for discussion: This was discussed in the original version of the manuscript. However, we felt that additional experiments could help further shed light as to whether the enhanced fatigue we observed in *P2ry1* mutants was mediated by perisynaptic events or by events along the entire muscle fiber. We reasoned that if direct muscle stimulation caused a similarly enhanced muscle fatigue in *P2ry1* mutants, then this would fail to support the idea that perisynaptic dysfunction, such as the failure to maintain potassium ionic homeostasis, was responsible for the effects on nerve stimulation-induced fatigue. We performed these experiments in muscle strips at P15, because we were unable to place the field stimulation electrodes at the appropriate positions to reproducibly cause fatigue in whole diaphragms at P7, similar to the setup used to produce fatigue in response to nerve stimulation. Moreover, we had performed nerve stimulation-induced fatigue in diaphragm strips at P15 and found an effect similar to that observed in P7 whole diaphragm (Figure 5—figure supplement 1). When we performed these experiments, we found that peak force and fatigue were not statistically different between *P2ry1* WT and mutant mice (peak force, 283.3+19.3 vs. 277+10.7 mN, *P2ry1* WT vs. mutant; *p*=0.32; ending vs. peak force, 93.2+2.1% vs.89.6+5.8%, *P2ry1* WT vs. mutant; *p*=0.19; *n*=3; see representative example in Author response image 3). In the revised manuscript, we added 2 sentences describing these experiments and described them as “data not shown.” We also added “the perisynaptic region of” in front of muscle fibers in the following sentence, in order to highlight this conceptual point of the finding. We also added a sentence in the Discussion that utilizes this data to support the idea that the fatigue-enhancing effects of high potassium are more likely to be caused by perisynaptic events rather than events along the entire fiber. Finally, we added a line about direct muscle stimulation in the Materials and methods.

**Author response image 3. respfig3:** 

Second item for discussion: The direct stimulation experiments argue against the idea that calcium signaling within TPSCs regulates the effects of potassium on muscle APs far outside the perisynaptic region. One way to measure the precise distance away from the endplate that is affected would be to record muscle APs at variable distances from this synaptic region during the latter stages of high-frequency nerve stimulation in *P2ry1* WT and mutant diaphragm. Muscle APs recorded in this manuscript were generated from synaptically positioned intracellular electrodes (e.g., in which the rise times of mEPPs were under 0.2 ms). We feel that while such an experiment is valid and would help address the precise distance away from the motor endplate that these effects could be achieved, that it is beyond the scope of this manuscript.

4) Revision of Figures:- Figure 1: are the sensory axons of the phrenic nerve recombined in Wnt-Cre mice (i.e. tdtomato+)? What exactly is depicted here (add to legend)?

Schwann cells along the phrenic nerve trunks and at the diaphragm NMJs are recombined in these mice. We added a sentence in the figure legend to describe this. The sensory afferents innervating muscle spindles (as well as other muscle sensory receptors) are neural crest-derived and thus also exhibit recombination. However, there are only a few spindles per rodent hemi- costal diaphragm (Barstad, Experientia, 1965 21(9):533-4) – the sensory innervation of one in the top left corner of the left panel in Figure 1 can be seen, in red.

- Figure 2: add significance.

Done, and ANOVA added to statistical methods.

- Figure 3: add variance and p values.

Done.

- Figure 4: consider changing the order (why is Figure 4 before 4C, D?);

Although we understand the rationale for showing the muscle APs first, since there is a phenotype, we feel that it makes more sense to show the EPPs first, simply because a.) they’ve been examined before b.) presynaptic precedes postsynaptic conceptually, and c.) their similarity between genotypes (4A) is conceptually similar to the similarity in percent AP success (4B-D). Only 4E shows a difference, which is the muscle AP shape. To place 4A after 4B would break up the AP figures (4B-E), and to place 4A after 4B-D would seem to be going backward from post to presynaptic.

- Figure 4: mention that the early up and downward deflections are stimulation artifacts;

Done, in figure legend.

- Figure 4: lacks quantification.

We’re assuming this was meant to be 5E, as 4E is quantified and there is no 4F or 5F. We’ve added the quantitative data to the figure legend.

- Figure 6: adding 10 mM [K^+^]_o_ had markedly different effects in muscle cells from wildtype and P2Y_1_-deficient mice. Can this be reproduced in wildtype muscle preincubated with MRS 21500?

We found similar effects on muscle fatigue exerted by high potassium in WT muscle treated with MRS2500. In addition, we found similarly reduced calcium responses to potassium in WT muscle treated with MRS2500. We have included this new data in Figure 6 (now called Figure 7) and the legend and mentioned it in the Results section.

- Figure 3—figure supplement 1 and Figure 3—figure supplement 2: the quantification of data is missing.

We’ve added the quantitative data to these 2 supplemental figure legends.

- Figure 6: did the authors also use P2Y_1_R mutant mice for calcium imaging at P15? It would be more consistent to include P2Y_1_R KO data in A/B. Preserved calcium signaling at P15 should be shown.

We found similarly absent calcium responses at P15 in *P2ry1* mutants as in MRS2500-treated *P2ry1* WT mice. We added this data to the figure. We also added a graph of these responses and included the quantitative data in the figure legend.